# Progress in Non-Traditional Processing for Fabricating Superhydrophobic Surfaces

**DOI:** 10.3390/mi12091003

**Published:** 2021-08-24

**Authors:** Dili Shen, Wuyi Ming, Xinggui Ren, Zhuobin Xie, Xuewen Liu

**Affiliations:** 1School of Mechanical-Electronic and Automobile Engineering, Zhengzhou Institute of Technology, Zhengzhou 450052, China; shendili@163.com; 2Mechanical and Electrical Engineering Institute, Zhengzhou University of Light Industry, Zhengzhou 450002, China; xzb18336947581@163.com; 3School of Vehicle and Automation, Guangzhou Huaxia Vocational College, Guangzhou 510900, China; liuxuewen@gzhxtc.edu.cn; 4Guangdong HUST Industrial Technology Research Institute, Guangdong Provincial Key Laboratory of Digital Manufacturing Equipment, Dongguan 523808, China

**Keywords:** superhydrophobic surfaces, laser beam machining, electrical discharge machining, electrochemical machining, ion beam machining, non-traditional processing, fabricating

## Abstract

When the water droplets are on some superhydrophobic surfaces, the surface only needs to be inclined at a very small angle to make the water droplets roll off. Hence, building a superhydrophobic surface on the material substrate, especially the metal substrate, can effectively alleviate the problems of its inability to resist corrosion and easy icing during use, and it can also give it special functions such as self-cleaning, lubrication, and drag reduction. Therefore, this study reviews and summarizes the development trends in the fabrication of superhydrophobic surface materials by non-traditional processing techniques. First, the principle of the superhydrophobic surfaces fabricated by laser beam machining (LBM) is introduced, and the machining performances of the LBM process, such as femtosecond laser, picosecond laser, and nanosecond laser, for fabricating the surfaces are compared and summarized. Second, the principle and the machining performances of the electrical discharge machining (EDM), for fabricating the superhydrophobic surfaces, are reviewed and compared, respectively. Third, the machining performances to fabricate the superhydrophobic surfaces by the electrochemical machining (ECM), including electrochemical oxidation process and electrochemical reduction process, are reviewed and grouped by materials fabricated. Lastly, other non-traditional machining processes for fabricating superhydrophobic surfaces, such as ultrasonic machining (USM), water jet machining (WJM), and plasma spraying machining (PSM), are compared and summarized. Moreover, the advantage and disadvantage of the above mentioned non-traditional machining processes are discussed. Thereafter, the prospect of non-traditional machining for fabricating the desired superhydrophobic surfaces is proposed.

## 1. Introduction

Superhydrophobic surface refers to the surface of a material or object that can make the water droplets falling on it exhibit a water contact angle (WCA) greater than 150° [1]. When the water droplets are on some superhydrophobic surfaces, the surfaces only need to be inclined at a very small angle to make the water droplets roll off. Such superhydrophobic surfaces have self-cleaning properties and can be used to prepare self-cleaning surfaces. In nature, the most typical example is the surface of the lotus leaf. In addition, bird feathers [2,3], water strider legs [4,5], and butterfly wings are all typical superhydrophobic surfaces. These surfaces all show the extremely difficult infiltration and wall-hanging of water in the macroscopic view. The reason for the superhydrophobicity is the special micro-scale topological structure presented on the surface. For example, water droplets gather into strands on the surface of the lotus leaf and flow down. The superhydrophobicity is mainly caused by its micron-sized papillary structure [6]. Figure 1 demonstrates the superhydrophobic surface of lotus leaf. The epidermis of the lotus leaf is composed of convex cells covered by wax tubes, with few stomata. The upper epidermis has a unique hierarchical structure consisting of papillae and dense wax tubules, which is the basis of the famous superhydrophobicity [7]. Figure 2 illustrates the typical wetting inherent in droplets (less than 0.5 μL) placed on pigeon feathers. [2]. The secret that the water strider can glide and jump gracefully at a maximum speed of about 600 km/h on the water is also in the special composite structure of its legs [4,5]. The needle-like bristles covering the surface of the legs of the water strider are about 50 μm in length, and the diameter gradually changes from 2 to 3 μm at the root to 100 nm at the tip, forming an inclination angle of about 20° with the surface of the leg. In addition, there are fine semi-helical nano-grooves on the surface of each bristle.

Young first established the Young’s model [8], according to which the surface of the material is smooth (ideally), the WCA between the liquid droplet (usually a water droplet) and the smooth material surface is constant, as shown in Figure 3a, and its size depends on the surface free energy. Barthlott [9] outlined more than 200 water-repellent plant species with WCAs greater than 150° and their surface morphologies. However, in the real world, absolutely smooth material surfaces do not exist. The actual material surface shows a certain degree of roughness, and the roughness has a very significant influence on the wettability. Therefore, Wenzel [10] improved Young’s model (Figure 3b), in which the liquid droplets completely fill the grooves on the rough surface. The model shows that due to the presence of the rough surface, the actual solid–liquid contact surface is larger than the area observed in the apparent geometry, thus enhancing the hydrophobicity. Furthermore, Cassie believed that there was air in the grooves of some rough surfaces, and liquid droplets could not fill the grooves, so the material surface should be composed of solid and gas (Figure 3c) [11]. The contact interface of liquid on the material surface included solid–liquid and gas–liquid contact interface. By increasing the gas–liquid contact area and reducing the proportion of solid–liquid contact area, a larger WCA could be obtained.

Due to its excellent water repellency, the superhydrophobic surface has a very broad application prospect in daily life and military equipment. The United States Ship (USS) McFall destroyer of the United States Navy has used protective outerwear made of superhydrophobic materials. This coat will protect the ship’s sensors, weapon systems, and other exposed equipment from salt spray corrosion. Among them, salt spray corrosion is the consequence of the oxidation reaction between the chloride ions contained in the chloride salt (the main component is NaCl) and the metal surface, which damages the surface structure of the metal and reduces the strength of the metal. This can significantly save time and money spent on ship maintenance [12]. In 2016, Rice University in the United States developed a graphene composite superhydrophobic material that can effectively prevent ice. When the temperature is higher than −14 °C, ice cannot condense on the surface of the material. Utilizing the conductive properties of graphene, the material can be electrically heated to prevent ice or deicing at lower temperatures, requiring only the application of 12 volts in order to make the material anti-icing at a low temperature of −51 °C [13]. China is also studying individual protective clothing based on superhydrophobic (superhydrophobic and super oleophobic) fabrics. Superhydrophobic/oleophobic fabrics can provide soldiers with up to 24 h of frost protection at −20 °C.

Therefore, building a superhydrophobic surface on the material substrate, especially the metal substrate, can effectively alleviate the problems of its inability to resist corrosion and easy icing during use; it can also give it special functions such as self-cleaning, oil-water separation, lubrication, and drag reduction. Fabrication of materials with superhydrophobicity have relatively high surface energy and exhibit obvious hydrophilicity, and it is relatively more difficult to prepare superhydrophobic surfaces with metal as a matrix than with glass and low surface energy polymers. Therefore, this study reviews and summarizes the development trends in the fabrication of superhydrophobic surface materials by non-traditional processing techniques, such as laser beam machining (LBM), electrical discharge machining (EDM), electrochemical machining (ECM), ultrasonic machining (USM), water jet machining (WJM), and plasma spraying machining (PSM).

## 2. Laser Beam Machining

### 2.1. Principle of the LBM

The LBM is a non-contact processing technology in which heat is transferred to the surface of the material through a high-energy laser beam, and the material is removed by melting, vaporization, and chemical decomposition in the local area irradiated by the laser spot. By changing the laser processing process parameters (energy density, scanning rate, scanning distance, laser frequency, and so on), the surface micro-structures with different morphology and size parameters can be obtained. Most lasers produce Gaussian beams, and the energy density is unevenly distributed in space. The energy density at the center is the highest, and the etching depth is the largest. The energy density becomes smaller and smaller as it moves away from the center line, and the etching depth gradually becomes shallow. According to different pulse times, lasers can be divided into femtoseconds, picoseconds, nanoseconds, and long pulse lasers. Except for femtosecond and picosecond lasers, materials are vaporized and rapidly cooled by generating heat in a very short time, and there is basically no thermal effect. Most of the other lasers show obvious thermal effects.

### 2.2. Machining Performances of the LBM

Wang et al. [14] studied the effect of ultra-short femtosecond laser pulses on the surface hydrophobicity of polymer polymethyl methacrylate (PMMA), and found that the change in surface wettability was highly dependent on the energy deposited on the surface of PMMA. Figure 4 plotted the WCA as a function of the energy deposition rate and the total energy deposited on the PMMA surface. At a lower laser fluence of 0.299 J/cm^2^, WCA increased and exceeded 90° while the total deposited energy accumulated was above 300 J/cm^2^. When the total energy increased to the range of 600–900 J/cm^2^, the WCA exceeded 120° (the maximum was about 125°). When the total energy was greater than 900 J/cm^2^, the WCA decreased with the increase in the total energy. In addition, Figure 4a,b also shows that the optimal energy deposition rate to achieve the maximum WCA was about 50 J/cm^2^/s during a single scan of the laser beam.

Lee et al. [15] used laser beam ablation technology to fabricate micro-pin arrays with a high aspect ratio on stainless steel (AISI 304). For vertical wall adhesion, the aspect ratio of the micro-pins should be high, long, and sharp. Because chromium oxide had a higher surface tension, the recast layer could accumulate to form micro-pins with a high aspect ratio. The micro-pins array was fabricated by the proposed method, and the result was shown in Figure 5. Using a 55 μm pitch multi-line scanning path, the peak power density was 2.5 kw/mm^2^. The scanning was repeated 2000 times, the penetration depth was 441.9 μm, and the length of the cast layer was 159.1 μm. In addition, the final micro-pins’ length was 601.0 μm, the tip spacing was 105 μm, and about 90 micro-pings could be fabricated. The fabricated micro-pins array revealed adhesion on the vertical wall and could produce 38.6 mN adhesion on the Ra 4.594 μm vertical wall, which could be successfully applied to the vertical wall attachment.

Xiao et al. [16] proposed a new method, based on liquid-phase laser ablation, to obtain a strong hydrophobic surface. In this method, a cemented carbide sample was immersed in a fluorosilane solution, the distance between the top surface of the sample and the liquid surface was controlled by laser ablation, and a surface with good hydrophobicity could be obtained. They studied the influence of processing parameters on the wettability of the machined surface (Figure 6 and Figure 7). As shown in Figure 6, the destruction of the hydrophobic structure led to a decrease in the WCA. Compared with the WCA before wear, the WCA after 600 times of wear at different frequencies was reduced by 5.4%, 11.2%, 6.9%, 22.1%, and 39%, respectively. This indicated that the smaller the laser frequency, the smaller the decrease in the WCA. In particular, when the frequency was between 20 and 80 kHz, as the number of wear increased, the WCA did not change significantly. This confirmed that the hydrophobic surface prepared in this frequency range was more stable and durable. As shown in Figure 7, the WCA at different scanning speeds was reduced by 11.7%, 13.4%, 20.9%, 22.8%, and 20.4%, respectively, compared with that of the unworn surface after 600 wear cycles [16]. The hydrophobic surface fabricated at a relatively low scanning speed was more stable than the surface fabricated at a higher scanning speed.

Xia et al. [17] used laser and silanization to fabricate the superhydrophobic surface of the micro-pillar array and simulated the impact of droplets on the superhydrophobic surface. In their study, nano-second laser texturing and FAS modification were used to fabricate the superhydrophobic aluminum surface. After laser treatment, the surface wettability changed from the original hydrophilic to super hydrophilic and then to superhydrophobic. In addition, by optimizing the simulation calculation grid and the fluid volume simulation method, the motion process of the droplet hitting the superhydrophobic surface was analyzed. Moreover, the morphological changes, internal pressure distribution and velocity distribution of the droplets were studied. The experiments confirmed that the simulation using the proposed method was in good agreement with the results. This can provide an accurate, low-cost, and meaningful reference for the selection of functional surface manufacturing methods in industrial applications. Sun et al. [18] used a femtosecond laser to control the hydrophobicity of the yttria-stabilized zirconia (YSZ) surface. Experimental results demonstrated that the laser texturing greatly improved the hydrophobicity of the YSZ surface, and the WCA increased from 86.7° to 151.8°. Cai et al. [19] fabricated superhydrophobic structures on 316 L stainless steel surfaces by nanosecond the LBM. They established a geometric model for laser processing of Gaussian micro-holes array and combined with constraints to predict and optimize the WCA. The results showed that when the laser power was 10 W or 14 W, with the increase in the micro-structure pitch, the WCA gradually increased, and then gradually decreased after reaching the peak. Validation experiments confirmed that the proposed model could predict the WCA and optimize the geometric parameters of the microstructure to achieve the maximum hydrophobicity. Zhang et al. [20] fabricated a gradient hydrophobic surface (GHS) on pyrolytic carbon by laser etching and fluorosilanization. Through the SEM measurement, it was found that the fabricated surface was composed of a bare pyrolytic carbon (PYC) area and four parallel ridges at different distances, with a certain gradient and showing hydrophobicity. The experimental results confirmed that a gradient hydrophobic surface was an effective way to obtain anti-thrombosis. Wang et al. [21] used a visible picosecond laser to enable single-step fabrication of glass micro/nanostructures for wettability control. In this way, a multi-level micro/nano structure could be fabricated in one step on any glass surface. This manufacturing capability enabled the design and manufacture of liquid-infused porous surfaces to prevent shear damage. Therefore, it can be used to quickly and effectively fabricate a liquid injection surface with complex micro-scale patterns and controllable nano-features. Table 1 draws the summary of the process of LBM described in the text grouped by laser type and object fabricated.

### 2.3. Summary

For laser processing of hydrophobic micro-structures, femtosecond lasers, picosecond lasers, and nanosecond lasers have all been successfully applied [14,15,16,17,18,19,20,21]. From the existing literature, nanosecond laser applications are the most common. This is because the price of nanosecond lasers are cheaper than femtosecond lasers or picosecond lasers, and the timescale of nanosecond lasers are shorter than femtosecond lasers or picosecond lasers. From the perspective of fabricating objects, it involves metals (carbide, stainless steel, aluminum-based alloys, and so on), ceramic materials, glass, and polymer materials. Among them, in the existing literature, there are more studies on metals, which is due to the wider use of them. In order to obtain a better hydrophobic surface, most of the research is to fabricate the array of grooves, holes, and other micro-structures through the rapid processing ability of laser. Some studies have also optimized its laser processing parameters to improve the micro-structure performance. Considering the thermal effect of nanosecond laser processing, most of the existing related studies still lack the analysis of the thermal effect of laser on micro-structures with hydrophobic properties, especially related theories and modeling analysis. Further, considering the commercial application prospects of hydrophobic micro-structure materials, it is also necessary to study the manufacturing cost of the hydrophobic micro-structure materials and make them faster and lower-cost in the future.

## 3. Electrical Discharge Machining

### 3.1. Principle of the EDM

In the electric discharge machining (EDM) process, the metal is melted at a high temperature under the action of the discharge current between the tool electrode and the surface and then solidifies on the surface to form a micro/nano structure. EDM uses the instantaneous high temperature generated when the spark between the tool electrode and the workpiece electrode is energized to remove the material on the surface of the workpiece. In recent years, the EDM method has been used to fabricate micro/nano structures with low surface energy and hydrophobicity on metal surfaces. By adjusting the process parameters, the micro/nano structure can be changed. This indicates that the process parameters can control the surface properties of the micro/nano structure, including surface energy and hydrophobicity.

### 3.2. Machining Performances of the EDM

Bae et al. [22] proposed a direct method of wire EDM (WEDM) to process micro-channels on the surface of stainless steel (AISI 304) to achieve wettability control. Figure 8 demonstrates the schematic illustration of the WEDM process and the machined sample. As depicted in Figure 8a, a commercial WEDM system was used to manufacture a micro-grooved wettability-controlled surface (electrode wire diameter: 250 μm). As shown in Figure 8b, of the manufactured sample, the sharpness of the corners of the ridges was relatively satisfactory. Since the processed electrode was a linear electrode (a circular cross-section), the corners of the groove had a circular shape. The experimental results showed that, through the WEDM process, a solid stainless steel surface with controllable wettability could be achieved. In order to enhance the hydrophobic and lipophilic properties, the depth of the microgrooves should be large enough. More importantly, the controllable wettability of stainless steel came from surface patterning without any additional chemical treatment on the machined surface, which is conducive to low-cost industrial application.

He et al. [23] investigated the anti-fouling effects of EDM machined hierarchical micro/nano structure (HMNS). Figure 9 depicted the relationship between the roughness (Ra and Rz) and the WCA of samples machined by the process of EDM. Among them, surface roughness Ra is the arithmetic mean deviation of the surface profile, the arithmetic mean of the absolute value of the profile deviation within the sampling length, and surface roughness Rz is the 10-point height of microscopic unevenness, the sum of the average of the 5 largest profile peak heights and the average of the 5 largest profile valley depths within the sampling length. As shown in Figure 9, with the increase in discharge current, more micro/nano structures were processed, resulting in an increase in roughness (Ra and Rz). However, with the further increase in the discharge current, the surfaces of samples 3 and 4 had more recast areas and molten balls than sample 2, resulting in a decrease in roughness (Ra and Rz). In addition, the WCA became larger as the discharge current increased. The experimental results showed that the modified surface of EDM had a hierarchical micro/nano structure. These micro/nano structures made the surface hydrophobic, thereby improving corrosion resistance and pollution resistance. With the increase in the discharge current in the process of EDM, the WCA increased. Therefore, this improved the anti-fouling performance and the fouling induction period of the hierarchical micro/nano structure.

Based on the EDM, Deng et al. [24] proposed a simple and scalable method for preparing CuO nanowires with V-shaped microgrooves by thermal oxidation. Figure 10 demonstrated the schematic of the manufacturing process for the V-shaped microgrooves. First, a low-cost EDM method was used to process V-shaped micro grooves on a copper plate (20 × 5 × 2 mm^3^). Subsequently, the processed micro-flute samples were ultrasonically cleaned with hydrochloric acid (HCl 10%), absolute ethanol, and deionized water to remove any residues or contaminants on the micro-flute. Then, it was dried with compressed air. Finally, the dried micro-grooved samples were put into a muffle furnace for thermal oxidation to prepare nanowires. The experimental results showed that, as the annealing temperature increased, the diameter of the nanowires monotonously increased, and, as the annealing time increased, the diameter of the nanowires remained almost unchanged. This showed that the annealing temperature had a positive effect on the length of the nanowires. All nanowire samples, which were fabricated by the proposed method, exhibited hydrophobicity (maximum WCA with 140°). Therefore, a low-cost manufacturing micro-slot radiator is expected to be applied in energy, micro-electronics, chemical, medical, and other fields.

Chen et al. [25] used the WEDM to fabricate a superhydrophobic structure on the surface of the SiCp/Al composite material. Figure 11 depicts the relationship between the designed parameters (radius and center distance) of the groove and the contact state. As depicted in Figure 11, the contact angles of the deep groove samples of (3) and (4) were basically equal to the original samples. In addition, when the semicircular groove had a sufficient height, the contact states of the droplets on both the samples (3) and (4) were in the Cassie–Baxter state. In addition, the experimental results showed that the maximum contact angle of the prepared microstructure (without other surface modification treatments) could reach 153.3°. This indicated that the surface of the composite material with superhydrophobic structure could be fabricated by the WEDM, and it had good wear resistance.

To obtain copper-based high-efficiency condensation heat transfer (CHT) interface, Chen et al. [26] fabricated a superhydrophobic hierarchical micro-groove with nanocone structure by the WEDM, electroless copper plating, and thiol modification technologies. The experimental results confirmed that the superhydrophobic hierarchical structure had the best performance. Compared with the flat hydrophobic copper, the CHT coefficient was increased by 82.9%. Based on magnetically aided electrode EDM (MAE-EDM), Xiao et al. [27] fabricated a superhydrophobic mesh. The experimental results showed that the processed textured mesh had good contact angle stability, corrosion resistance, and mechanical stability. Peta et al. [28] investigated the relationship between the contact angle and surface topographies created by the EDM process. The experimental results showed that the wetting behavior of aluminum alloy could be easily controlled by appropriately adjusting the process parameters of the EDM. Wu et al. [29] studied the effect of sub-millimeter morphologies, with a sinusoidal structure or a rectangular one, on the hydrophobicity of a copper surface, which was machined by the WEDM process. The experimental results showed that the servo voltage had a significant effect on the hydrophobicity of the copper surface, and the sinusoidal structure had better hydrophobicity than the rectangular one. In addition, the combination of the EDM and other machining methods was also used in order to fabricate micro-structure with good hydrophobicity [30,31,32]. Table 2 demonstrates the summary of the process of EDM described in the text grouped by machining type and object fabricated.

### 3.3. Summary

According to the existing literatures, it is feasible for EDM to cut a hydrophobic micro-structured surface, such as grooves, micro/nano structure, and hydrophobic meshes; the processing steps of EDM are few and it is convenient for large-scale industrial applications in the future. Generally speaking, the machining quality (precision and surface quality) of EDM is in conflict with the material removal rate (MRR) [33,34,35]. Therefore, under the premise of ensuring that the hydrophobic micro-structured surface can be completed, it is necessary to study the improvement of MRR and optimize its machining performances to further reduce the cost of large-scale manufacturing in the future. Limited by the principle of EDM, most of the existing machining objects are conductive materials (as shown in Table 2), such as metals, conductive composite materials, or conductive ceramic materials. However, non-conductive materials (glass and non-conductive ceramic materials) are also very popular in industrial applications. Therefore, in the future, methods such as auxiliary electrode EDM or high-voltage discharge EDM could be used to solve this problem [34,36,37].

## 4. Electrochemical Machining

### 4.1. Principle of the ECM

ECM is a method that uses electrochemical reaction (or electrochemical corrosion) to process metal materials [38,39,40]. Compared with mechanical processing, electrochemical processing is not limited by material hardness and toughness. Therefore, many metal materials can be machined by the ECM process regardless of their hardness. However, its machining accuracy, surface roughness, and other surface qualities are one of the obstacles restricting the popularization and application of this technology. In addition, surface characteristics such as surface roughness and wettability become very important when finishing with electrochemically machined workpieces. According to the existing literatures, the fabrication of hydrophobic surfaces mainly adopts electrochemical oxidation process and electrochemical reduction process (metal and metal oxide deposition) in the ECM process. This reaction produces metal ions through metal oxidation in the presence of protons and then reacts with water to form metal oxides [41]. Moreover, electrochemical reduction of metal ions is a widely used deposition method for various metals.

### 4.2. Machining Performances of the ECM

#### 4.2.1. Electrochemical Oxidation Process

**Aluminum:** Generally, anodization of aluminum can be used to create closed hexagonal nanopores. By adjusting the electrochemical parameters, the length, diameter, and pore spacing of the nanopores can be controlled. Since the acid resistance of the nanopore wall is lower than that of the upper surface, anodized aluminum oxide can undergo anisotropic corrosion after the anodizing step. To obtain poly (vinyl alcohol) nanofibers, Feng et al. [42] utilized the anodized aluminum surfaces as template. By extrusion of the polymer inside the template, a superhydrophobic surface with the WCA of 171.2° could be achieved. Zhang et al. [43] used an electrolytic oxidation process and a porous anodic aluminum oxide film as a template to prepare a two-dimensional array of perfluoropolyether derivative nanopillars. The experimental results showed that the nanopillars on the lotus leaf-like topology had low contact angles, thus exhibiting superhydrophobicity and self-cleaning properties. Neto et al. [44] fabricated a surface formed by densely arranged nickel nanowires (nanocarpet) by electrodepositing the aluminum oxide film template and then dissolving the film. The nickel nanowires that form the nanocarpet had a very high aspect ratio (about 250), a diameter of 200 nanometers, and a length of tens of microns. Once coated with a hydrophobic surfactant (stearic acid), the nickel nanocarpet had superhydrophobicity (WCA = 158°) and maintained its superhydrophobicity after being immersed in water for a period of time. Savio et al. [45] studied the direct relationship between the hydrophobicity of the micro-structured aluminum surface and the surface chemistry. They found that the corroded aluminum surface had a binary structure of nano-scale block bumps and depressions, which provided more space for air trapping. The excellent hydrophobic property of aluminum surface was due to its dual structure of layered micro/nano structure. Experiments showed that the WCA was indeed inversely proportional to the content of surface metal aluminum, and the hydrophobic behavior depended on the combined effect of surface morphology and surface chemistry.

**Silicon:** In an electrolyte containing fluoride ions, anodizing of silicon is a feasible method to form a porous film with high porosity. By integrating with active electronic devices, for example, the silicon-based superhydrophobic surface can protect these devices from the harmful effects of environmental water and moisture, making relevant research attractive. Wang et al. [46] introduced a simple and inexpensive method that combined electrochemical surface modification (to produce micro/nano topography with fractal shapes) and wettest etching steps to fabricate superhydrophobic silicon surfaces with the WCA of 160°. Balucani et al. [47] proposed a method of making superhydrophobic surfaces with porous silicon. They used electrochemical methods to modify the morphology of p-type silicon and obtained porous materials with controllable size and distribution. In addition, large contact angles were observed on these surfaces, which had superhydrophobic properties.

**Titanium**: In an electrolyte containing fluoride ions, titanium can be anodized by an electrochemical processing method to obtain fine titanium oxide nanotubes. During the formation of the nanotubes, fluoride ions play a leading role in the dissolution of TiO_2_. Wang et al. [48] combined the ECM (electrochemical oxidation process) with the LBM to create superhydrophobic surfaces on titanium materials. The method basically involved first forming a micro-structure on the Ti surface, forming TiO_2_ nanotubes on it, and then post-modifying the hydrophobic material. The experimental results showed that, through different methods, the control of wettability could be achieved on these surfaces.

**Copper**: By anodic oxidation of copper in KOH or NaOH aqueous solution, it is possible to form copper hydroxide nano-needles. Wu et al. [49] reported on the preparation and characterization of stable superhydrophobic surfaces. The thin copper layer was anodized to form a copper hydroxide nano-needle film, which then reacted with n-dodecyl mercaptan to form a thermally stable Cu(SC_12_H_25_)_2_ superhydrophobic coating. The experimental results showed that the contact angle of the modified nano-needle surface was greater than 150°, and the inclination angle was less than 2°. In addition, when it heated in the air for more than 42 h (over 160 °C), the surface of the copper still maintained its superhydrophobic properties.

**Others**: He et al. [50] used electrochemical anodization to fabricate a ZnO film on zinc foil, which had a variety of structures such as nano-dots, nano-wires, and nano-flowers. Experiments confirmed that, under the action of a DC or AC electric field, the surface of the ZnO film with electro wettability changed from a hydrophilic state to a superhydrophobic state. Shu et al. [51] reported the use of self-organized anodic oxidation to make a superhydrophobic surface with a hierarchical structure. They changed the size of the micro-capsules and the tip angle by adjusting the potential difference and the water content in the electrolyte. In addition, the decrease in water content increased the size of the micro-capsules, and the nanofibers became nanoparticles. By coating a layer of fluoroalkyl phosphate on the anodized film, they found that the surface was superhydrophobic, with a WCA of up to 175°.

Therefore, anodizing is a widely used technology that uses electrochemical cells to artificially generate an oxide layer on the surface of non-precious metals. The Summary of electrochemical oxidation process described in the text grouped by materials fabricated is listed in Table 3.

#### 4.2.2. Electrochemical Reduction Process

**Gold**: Due to its unique optical, electrical and chemical properties, gold micro/nano structures have attracted much attention for their applications in catalysis, sensing, biomedicine, and self-cleaning. Ye et al. [52] directly deposited gold micro-structures (hierarchical dendrites with secondary and tertiary branches) on the indium tin oxide (ITO) substrate by the ECM method without using any template, surfactant, or stabilizer. They studied in detail the influence of the potential in the ECM and the concentration of HAuCl_4_ on the formation of the deposition process. Compared with the bulk gold electrode, the synthesized surface showed higher electro catalytic activity and stability to the electro oxidation of ethanol. Even if the surface after ECM processing was not modified by coating, it still had hydrophobicity, and it had significant superhydrophobicity even in corrosive solutions with a wide pH range. Shi et al. [53] fabricated superhydrophobic coatings on gold wires by combining the ECM with the layer by layer (LBL) technique. The experimental results showed that a small amount of micro/nano gold aggregates were formed on the surface of gold wire. The results also showed that the superhydrophobic coating with micro/nano gold aggregates could provide more bending force. This understanding opens up new application prospects for bionic drag reduction and rapid advancement technology. Ren et al. [54] proposed a two-step ECM method for preparing hierarchical cauliflower-like gold structures. The experimental results demonstrated that after being modified by fluoroalkylsilane, the surface of the cauliflower-like structure presented a high contact angle (WCA = 161.9°) and a low sliding angle, reaching a superhydrophobic surface. Shepherd et al. [55] used electrochemical methods to prepare heterogeneously mixed monolayers on the surface of polycrystalline gold. Experiments showed that a local hydrophobic environment was formed near the molecular membrane.

**Silver**: Using electrochemical deposition, the micro-structures of silver polyhedrons and silver dendrites can be prepared on a copper substrate covered with a nickel film. Zhao et al. [56] proposed a new method of electrochemical deposition of Ag aggregate dendritic structure on the polyelectrolyte multilayer film substrate through the ECM processing method. The morphology of silver aggregates could be adjusted by ECM process parameters, such as electrochemical deposition time and potential. In addition, after chemisorption of n-dodecanethiol monomolecular film, the prepared WCA was as high as 154°, and it became superhydrophobic. Gu et al. [57] adopted electrochemical methods to grow single crystal Ag dendrites on Ni/Cu substrates. Similarly, by controlling the potential process parameters in the ECM, the morphology of the deposited silver could be transformed from polyhedrons to dendrites. The micro-structure characterization showed that, through directional attachment, silver dendrites (composed of tree trunks, branches, and leaves) could be formed. After modifying the silver dendritic micro-structure with a monomolecular film of about 10 μm, a superhydrophobic surface with a WCA of 154.5°+/−1.0° and an inclination angle of less than 2° could be obtained.

**Copper and Copper Oxides**: Since copper is naturally oxidized when it comes in contact with air, copper oxide is formed on its surface. By adopting a constant potential electrochemical method, a spherical copper micro-structure with a hierarchical structure can be prepared. Huang et al. [58] fabricated a particulate superhydrophobic aluminum surface by electrochemical depositing copper on the aluminum surface and then electrochemically modifying it. Experiments showed that the density of its particulate micro-structure decreased with the increase in deposition potential. When the electrochemical potential increased from 0.2 to 0.8 V, the particle spacing decreased from 26.6 to 11.03 μm. Therefore, the aluminum substrate had a superhydrophobic surface with a roughness of 6–7 μm (WCA = 157°).

**Si and Ag +**: If the electrolyte contains fluoride ions, the silicon substrate can also be used for electrochemical deposition. In this process, silicon corrosion and metal deposition occur simultaneously. Yang et al. [59] proposed a method that combined deep reactive ion etching and photolithography to fabricate a silicon micro/nano layered structure. By adjusting the process parameters, the morphology of the nanostructures could be partially controlled, thereby exerting a controllable influence on the surface properties. The contact angle measurement results confirmed that the prepared silicon surface with a layered structure had superhydrophobicity.

**Others**: Huang et al. [60] proposed an electrochemically deposited composite coating with a thorn-like hierarchical structure with high roughness to obtain a superhydrophobic surface. They found that, by adjusting the process parameters such as current density and electro deposition time, the geometry of this hierarchical structure could be controlled to make the contact angle as high as 174.9°. Importantly, this method could be easily extended to other conductive materials. Because this proposed method saves time and money, it has broad application prospects in the industrial field. Xue et al. used [61] an electrochemical deposition method to fabricate a bimetallic NiCo coating with a layered micro-sphere structure on a carbon steel substrate. After modification with low surface energy materials, the layered micro-sphere structure of NiCo coating had a very high contact angle (about 165°) and exhibited superhydrophobic properties. Therefore, it has good anti-corrosion performance for bare carbon steel. In addition, Wang et al. [62] fabricated tungsten carbide particles reinforced Co-Ni superhydrophobic composite coatings by electrochemical deposition. The tests showed that the prepared superhydrophobic Co-Ni/WC composite coating (with a WC content of 9.8 wt %) had excellent wear resistance.

In the above methods, metal oxides can be formed on the outermost surface, and their morphology largely depends on electrochemical parameters. The summary of electrochemical reduction process described in the text grouped by materials fabricated is listed in Table 4.

### 4.3. Summary

In the electrochemical oxidation process, compared with other conductive materials, aluminum has the highest frequency as a research object. On the one hand, this aspect is related to the physical and chemical properties of aluminum; on the other hand, it is also related to its widespread use. In addition, difficult-to-process materials such as titanium and semiconductor silicon are also common machining objects. These materials are related to sealing components, micro-electromechanical systems, and so on. In the electrochemical reduction process, the precious metals, gold and silver, are more frequently studied as deposited materials. This is because their chemical properties are relatively stable and their physical properties are very good, and they have irreplaceable roles in industrial applications. For example, in Table 4, the WCA of gold materials after electrochemical deposition is significantly greater than that of silver or copper materials. Whether it is electrochemical oxidation process or electrochemical oxidation process, its processing technology is simple and economical. Therefore, electrochemical machining of the micro-structured surface has a better application prospect.

## 5. Other Non-Traditional Machining

### 5.1. USM

Ultrasonic machining has developed rapidly in the past decades, and it is not limited by whether the material is conductive or not. The macroscopic force of the tool on the workpiece is small, as is the thermal effect. Therefore, in the field of difficult-to-process materials, many key process problems have been solved, and good results have been achieved [63,64,65,66]. Chen et al. [67] used ultrasonic machining to treat the surface of polycarbonate material after laser processing, and studied its surface wettability. Experimental results showed that after ultrasonic treatment, the superhydrophobic surface and hydrophobic properties were reduced, but the surface remained hydrophobic. In order to obtain the gas diffusion layer of the proton exchange membrane fuel cell, Guo et al. [68] constructed a hydrophobic and hydrophilic synergistic surface with ultrasonic atomization spray technology. Liang et al. [69] used micro-ultrasonic powder molding to fabricate micro-plastic parts with hydrophobic surfaces. The experimental results showed that the micro-plastic parts had good surface hydrophobicity (WCA = 135.4°). Zhao et al. [70] reported an ultrasonic vibration-assisted laser composite machining, depicted in Figure 12, to fabricate superhydrophobic copper surfaces. The experiments showed that the WCA of the superhydrophobic copper surface was 157.4° (under 3 μL water). Chen et al. [71] used rubber bands (RB) and low-cost carbon black nanoparticles (CBNPs) as polymer matrix and conductive nano fillers to prepare conductive polymer composite with superhydrophobic for wearable strain sensors. Under the action of ultrasound, the CBNPs were evenly distributed on the surface of RB.

### 5.2. WJM

The WJM process uses the principle of liquid pressurization to convert the mechanical energy of the power source (motor) into pressure energy through a specific device (pressurization port or high-pressure pump). The water with huge pressure energy is passed through the small hole nozzle (another change energy device), and then converts the pressure energy into kinetic energy to form a high-speed jet [72,73,74,75,76]. Shi et al. [77] studied the superhydrophobic texture of metal surfaces machined by water jet guided laser (WJGL). Figure 13 demonstrates the schematic diagram of the WJGL process and machined results by them. It could be clearly seen from Figure 13b that the heat-affected zone when the water jet guided the laser processing was very small. Validation experiments confirmed that the contact angles on the surface of 304 stainless steel, titanium, and 6061 aluminum stabilized at 150°, 130°, and 129°, respectively, after the water droplets on the machined micro-structured surface were placed for 20 days.

Yoshitaka et al. [78] combined the photoresist mask process and the micro-fluid jet process to mechanically remove the glass material and studied the effect of its surface micro-structure. The relationship between the process parameters and the processing accuracy and the hydrophobic/hydrophilic properties of the processed surface was evaluated. Furthermore, Yang et al. [79] studied the collision process between Newtonian fluid jets and hydrophilic and hydrophobic surfaces based on the impinging jet theory. The results showed that, with deionized water as the experimental working fluid, the theoretical calculation results were in good agreement with the experimental data, and the hydrophobicity of the surface had a significant effect on the size of the liquid phase sheet on the solid surface.

### 5.3. PSM

Plasma spraying machining is a thermal spraying method that uses a non-transferred plasma arc as the heat source, and the spraying material is powder [80,81]. In the past ten years, plasma spraying technology has developed rapidly. At present, atmospheric plasma spraying, controlled atmosphere plasma spraying, solution plasma spraying, and other spraying technologies have been developed. Plasma spraying has become the most important process method in thermal spraying technology [82,83]. Sohbatzadeh et al. [84] used a combination of electrospray and atmospheric pressure argon plasma jet, depicted in Figure 14, to deposit a hydrophobic diamond-like film on the surface of cotton fabric. The results showed that the WCA of the prepared sample increased after the combination process. In addition, the longer the coating time, the better the anti-aging effect. In addition, the hydrophobic properties of the samples had acceptable durability after 50 washing cycles. Ting et al. [85] applied an atmospheric plasma jet to prepare a hydrophobic coating on the glass surface. The experimental results demonstrated that the WCA was 105.7° under the best discharge parameters, and that the hydrophobic properties of the coated glass after preparation remained stable after 50 days. Asadollahi et al. [86] deposited an organosilicon-based superhydrophobic coating on micro-roughened Al-6061 substrate through an atmospheric pressure plasma jet. The results revealed that multi-pass depositions changed the surface morphology and surface chemistry, and affected the hydrophobicity of the coating. In addition, the increase in coating thickness improved the stability of the coating under corrosive conditions. However, the increase in oxygen content reduced the hydrophobicity of the coating.

### 5.4. Summary

It can be known from the above-mentioned literatures that the USM process and the WJM process are generally combined with other non-traditional machining techniques to fabricate superhydrophobic surfaces that meet the requirements. For example, the combination of ultrasonic processing and laser processing can improve the micro-scopic morphology of the micro-structure after processing [70]. Additionally, the combination of water jet processing and laser processing can reduce the size of the heat-affected area of the laser and enhance the processing accuracy [77]. However, the existing related mechanisms and models for ultrasonic machining of superhydrophobic surfaces are still relatively lacking. For the PSM process, it can fabricate superhydrophobic surfaces not only on metallic materials, but also on non-metallic materials. However, the PSM process has certain requirements on the heat resistance of the material, and the manufacturing cost is relatively high. Therefore, studies on the performances of the PSM process to prepare superhydrophobic surfaces also need to consider the economics of large-scale mass production.

## 6. Discussion

Non-traditional processing generally refers to machining methods that use electrical energy, thermal energy, light energy, electrochemical energy, chemical energy, sound energy, or special mechanical energy to remove or increase materials, so as to achieve material removal, deformation, performance change, or plating cover. Femtosecond laser and picosecond processing have no obvious thermal effect, but the equipment investment cost is high. The investment cost of nanosecond processing equipment is low, but there is obvious thermal effect during processing [14,15]. EDM melts and vaporizes materials through the high temperature of the discharge channel, and has the outstanding ability of difficult-to-process materials. However, the electrode consumption is large for the processing of the micro-hole arrays. Due to the limitation of the wire profile, the bottom of the micro-channel processed by WEDM cannot be processed into a “V” shape [25]. In addition, for processing super-hydrophobic surfaces, EDM is mainly oriented to conductive metal materials. To potentially fabricate high-performance super hydrophobic surface, Zhang et al. [87] proposed a novel micro machining technique, magnetically controlled ultrashort pulsed laser induced plasma micro-machining. Kliuev et al. [88] compared the microstructure of the γ-TiAl surface processed by WEDM and die-sinking EDM, and found that these processes can be used as industrial solutions for the generation of superhydrophobic and superhydrophilic surfaces. Due to its impact on the environment, electrochemical machining [89,90] is considered to be a non-traditional machining method that is not friendly to the environment, requiring the recovery and harmless treatment of the waste electrolyte. In the WJM process, due to the certain fluctuations in the width of the cut in the water jet processing, it is necessary to use micro-fluids or combine with other non-traditional processing methods. Other non-contact processes, such as the USM and the PSM processes, also have their advantages and disadvantages, as reported by different researchers [91]. Considering the large-scale manufacturing of super-hydrophobic surfaces, the economics of all of the above-mentioned processing methods need to be discussed. Table 5 shows the relative comparisons of different non-traditional machining processes, mentioned above, for fabricating superhydrophobic surfaces.

Figure 15 compares the machining efficiency, machining accuracy, machining cost, and environmental impact of different non-traditional machining methods for superhydrophobic surfaces. For example, the cost of a picosecond laser is significantly better than that of a femtosecond laser, and the production efficiency of picosecond laser processing is higher than that of a femtosecond laser. Moreover, the machined hydrophobic micro-structure is stable, and the shape is controllable by the picosecond laser. However, the disadvantage is still that the economy is not ideal, and it does not have a cost advantage in large-scale production. In recent years, nanosecond laser processing technology has attracted much attention due to its efficient processing speed. However, the thermal effect of nanosecond laser processing will cause many difficult problems such as recast layers, cracks, and self-organized structures, which will seriously affect the quality and performance of the fabricated superhydrophobic surface. In addition, compared with other machining methods, the EDM process is a green machining method with less pollution and low cost, which has the potential to prepare surfaces with excellent antifouling performance [92,93,94].

## 7. Outlook

Constructing a superhydrophobic surface can effectively alleviate the problem of its inability to resist corrosion and easy freezing during use [95,96]. It can also be given special functions such as self-cleaning, lubrication, and drag reduction. Using existing non-traditional machining technologies, such as the processes of LBM, EDM, and ECM, superhydrophobic surfaces can be manufactured. The development prospects of non-traditional machining for fabricating superhydrophobic surfaces are prospected as follows:For laser processing, femtosecond laser and picosecond laser have relatively little thermal effect on material removal processing. However, compared with nanosecond laser, their equipment purchase costs are high, and the machining costs are also high. For the manufacture of superhydrophobic micro-structures, femtosecond laser and picosecond laser methods must consider their enthusiasm and reduce manufacturing costs. In the future, manufacturing costs need to be considered as a constraint or an optimization goal.As the EDM process is restricted by its technology [97], it is difficult to fabricate surfaces with superhydrophobic micro-structures from non-conductive materials (glass and non-conductive ceramic materials). In the future, a breakthrough in the EDM technology should be addressed. In addition, special superhydrophobic micro-structures with good preparation properties should combine the EDM process with other machining method, such as laser processing and ultrasonic machining methods.When the ECM process is used to manufacture superhydrophobic surfaces, many researches mainly use template-free processing technique, which is simple and economical. However, the template-free electrochemical machining method is difficult for micro-channels, array micro-holes, and other surfaces with high-precision requirements. It is necessary to prepare high-precision electrodes and to overcome most of the problems related to processing technology. Therefore, the cost of large-scale manufacturing of superhydrophobic materials can be further reduced.It is not very common for the USM or WJM processes to independently fabricate superhydrophobic surfaces. It is necessary to further expand the study of related mechanisms/processes and establish related simulation models to improve the quality of machining process. The machining mechanism of USM or WJM processes combined with other non-traditional machining methods also needs to be elaborated through the establishment of related models. In addition, the machining economy of the PSM process also needs attention in future research.Due to the particularity of non-traditional processing, some processing techniques have a negative impact on the environment. For example, the harmless treatment of waste electrolyte in electrochemical machining, and the toxic gas of oily dielectric in EDM machining. For the future manufacturing of large-scale superhydrophobic materials, not only processing time and processing costs, but also environmental impacts need to be considered, such as the use of environmentally friendly electrolytes, optimization of laser processing energy, and so on.

## Figures and Tables

**Figure 1 micromachines-12-01003-f001:**
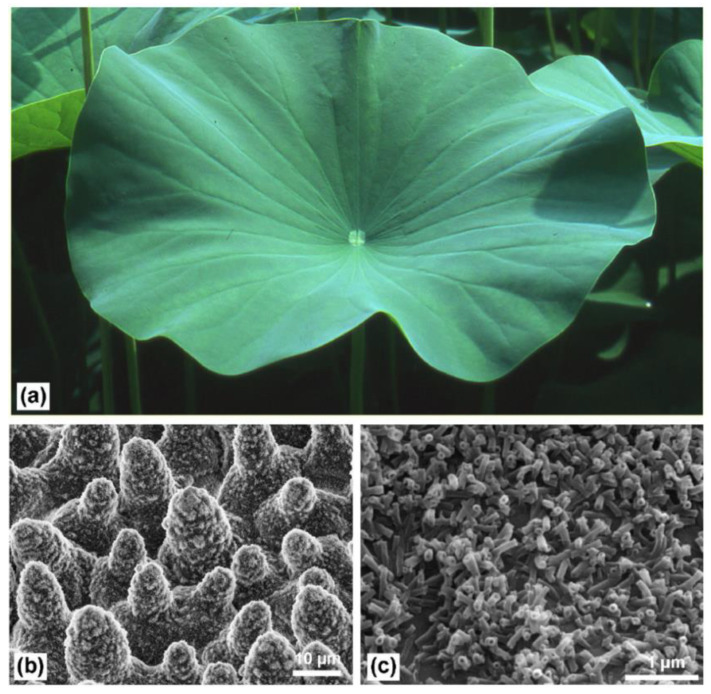
The superhydrophobic surface of lotus leaf; (**a**) general image of lotus leaf; (**b**) SEM image of the upper leaf side (a hierarchical surface structure consisting of papillae, wax clusters and wax tubules); (**c**) SEM image of wax clusters on the upper leaf side [7].

**Figure 2 micromachines-12-01003-f002:**
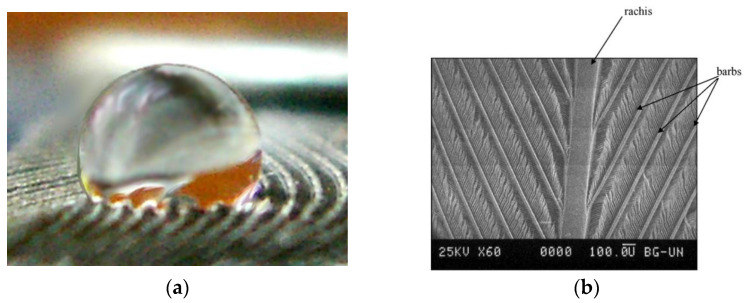
The superhydrophobic surface of bird feathers [2]; (**a**) general image of water droplets placed on bird feathers; (**b**) SEM image of the central part of a bird feather. Scale bar is 100 µm.

**Figure 3 micromachines-12-01003-f003:**
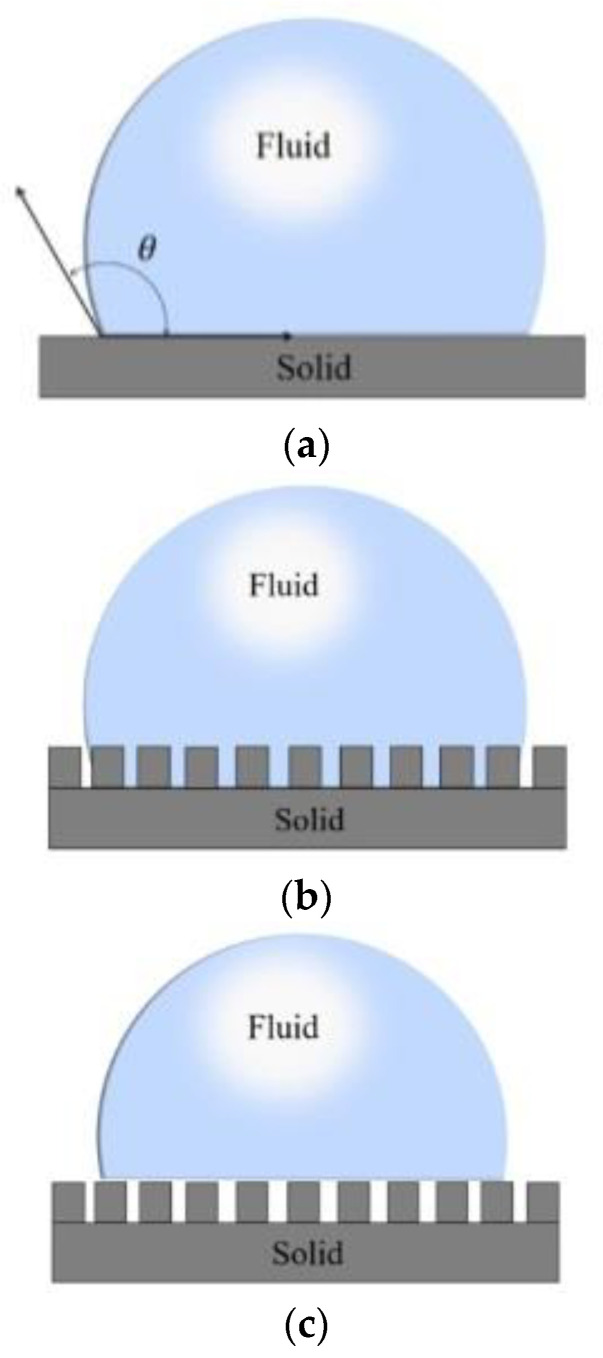
Different wettability models; (**a**) the Young model; (**b**) the Wenzel model; (**c**) the Cassie model.

**Figure 4 micromachines-12-01003-f004:**
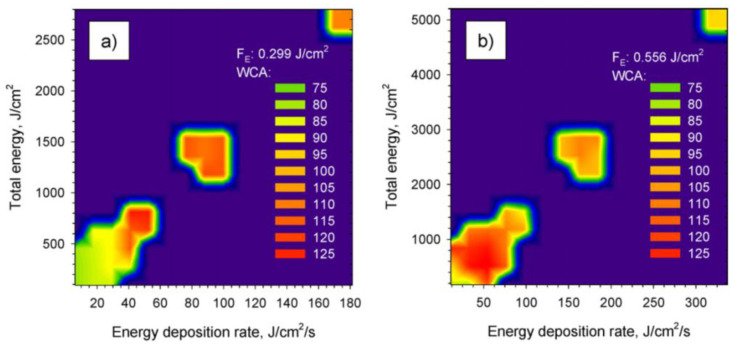
Effect of energy deposition rate and total energy on the WCA at different laser fluences [14]; (**a**) 0.299 J/cm^2^; (**b**) 0.556 J/cm^2^.

**Figure 5 micromachines-12-01003-f005:**
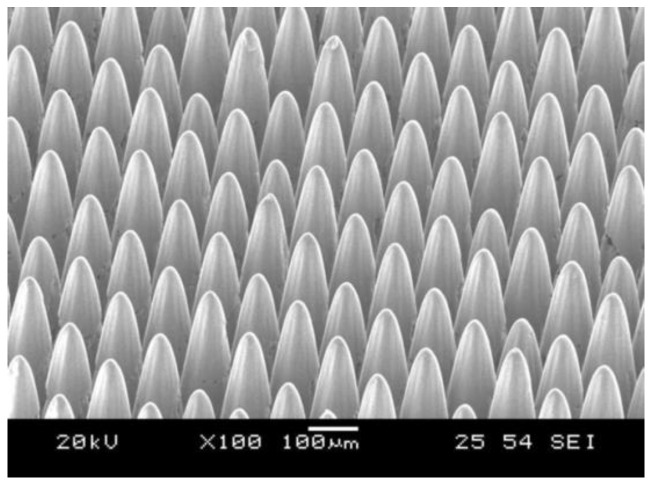
Micro-pins array fabricated on stainless steel sheet (AISI 304, 3 mm × 3 mm) by the process of LBM [15].

**Figure 6 micromachines-12-01003-f006:**
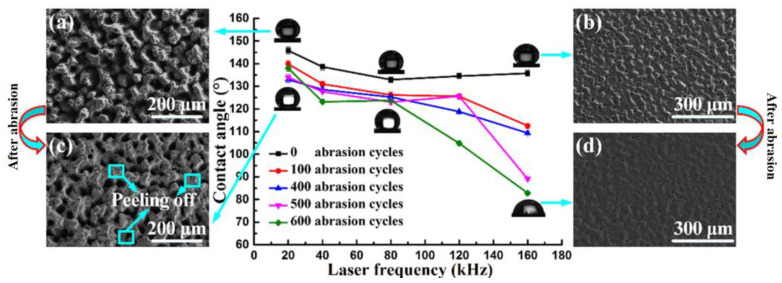
The influence of laser frequency on wettability (SEM images) [16]; (**a**) the hydrophobic surface fabricated at 20 kHz before abrasion; (**b**) the hydrophobic surface fabricated at 160 kHz before abrasion; (**c**) the hydrophobic surface fabricated at 20 kHz after 600 abrasion cycles; (**d**) the hydrophobic surface fabricated at 160 kHz after 600 abrasion cycles.

**Figure 7 micromachines-12-01003-f007:**
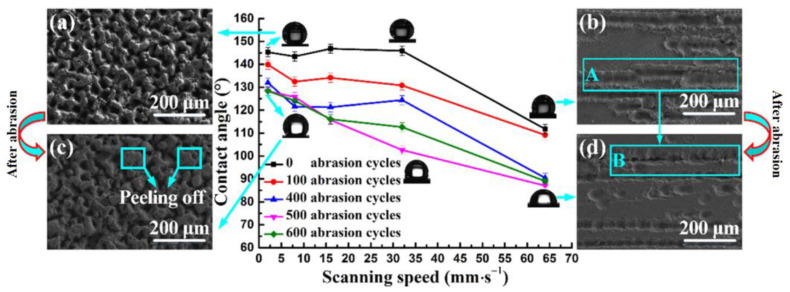
The influence of laser scanning speed on wettability (SEM images) [16]; (**a**) the hydrophobic surface fabricated at 2 mm/s before abrasion; (**b**) the hydrophobic surface fabricated at 64 mm/s before abrasion; (**c**) the hydrophobic surface fabricated at 2 mm/s after 600 abrasion cycles; (**d**) the hydrophobic surface fabricated at 64 mm/s after 600 abrasion cycles.

**Figure 8 micromachines-12-01003-f008:**
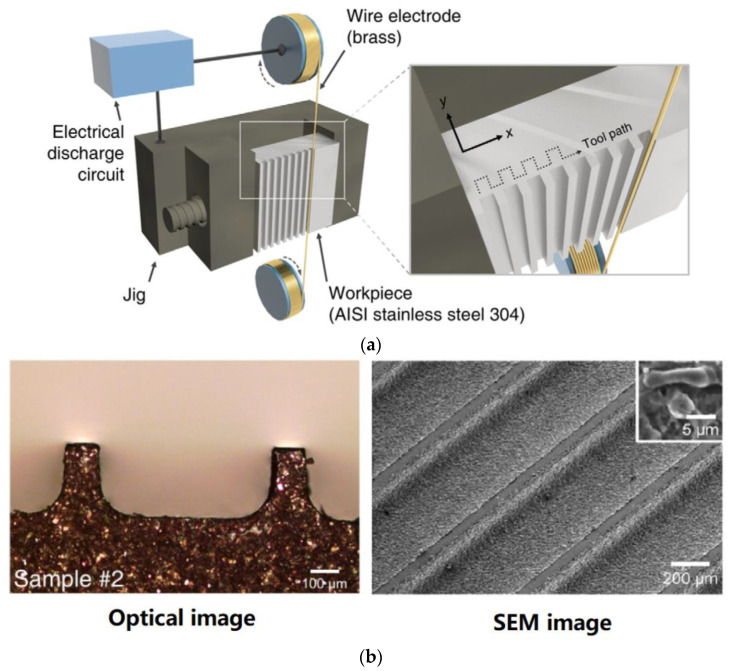
Schematic of the WEDM process and the machined sample [22]; (**a**) schematic illustration of the process; (**b**) optical image and SEM image of the sample.

**Figure 9 micromachines-12-01003-f009:**
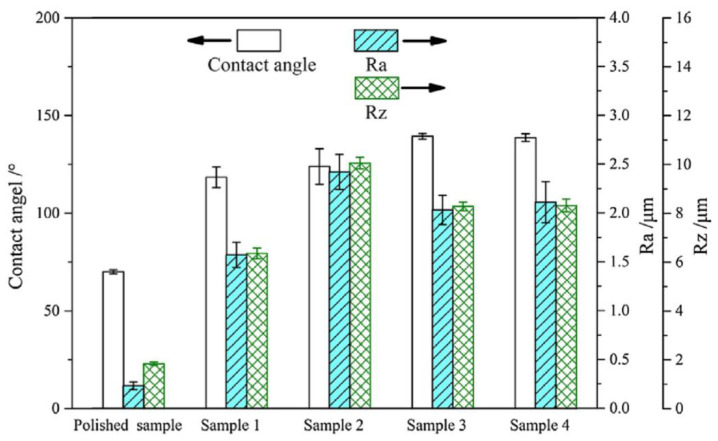
Relationship between the roughness (Ra and Rz) and the WCA of samples machined by the process of EDM [23].

**Figure 10 micromachines-12-01003-f010:**
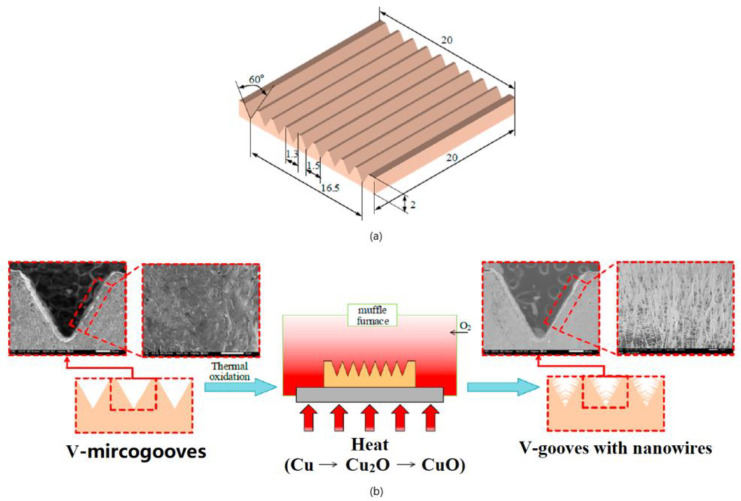
Schematic of the manufacturing process for the V-shaped microgrooves; (**a**) geometric dimensions; (**b**) schematic of the thermal oxidation process [24].

**Figure 11 micromachines-12-01003-f011:**
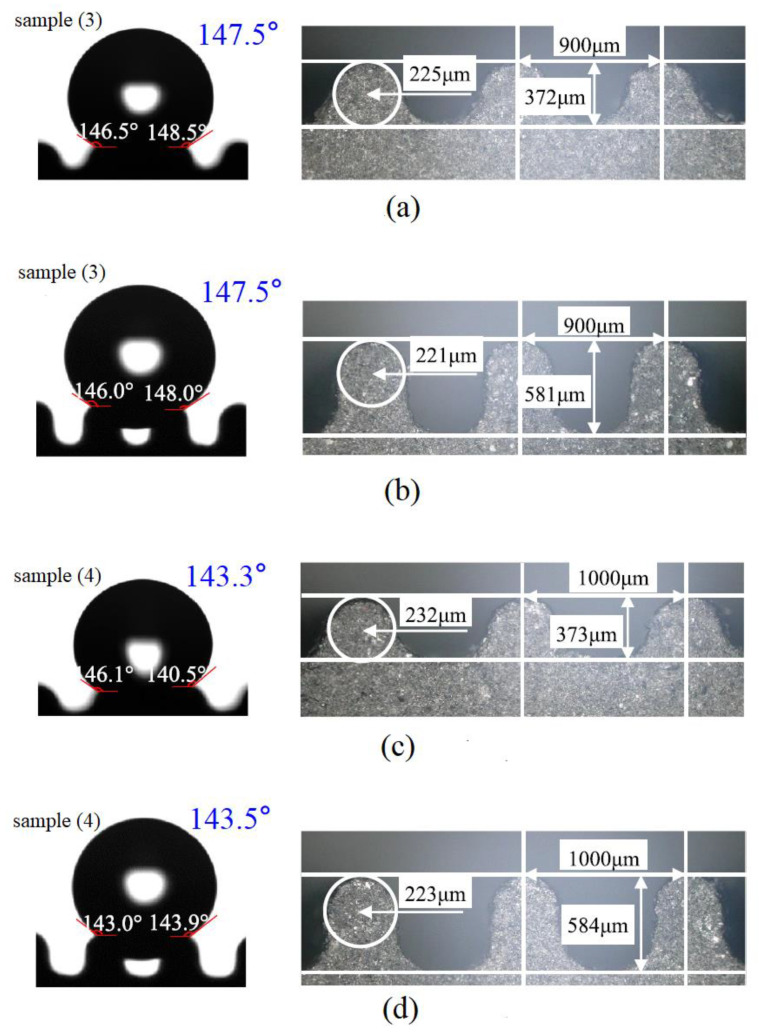
Relationship between designed parameters of the groove and the contact state [25]. (**a**) The original sample (3); (**b**) the sample (3) with deep groove; (**c**) the original sample (4); (**d**) the sample (4) with deep groove.

**Figure 12 micromachines-12-01003-f012:**
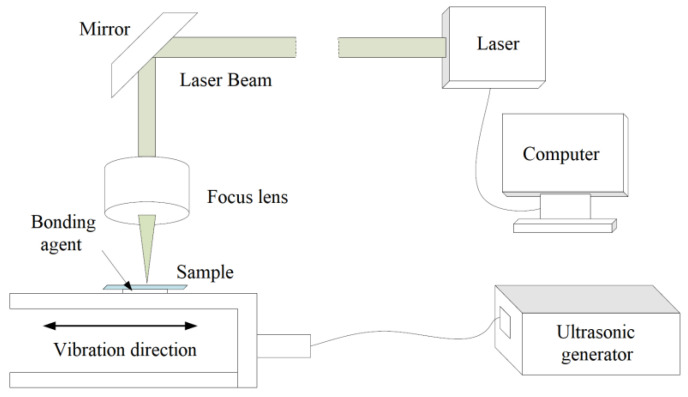
Schematic of ultrasonic vibration-assisted laser composite machining of copper for superhydrophobicity [70].

**Figure 13 micromachines-12-01003-f013:**
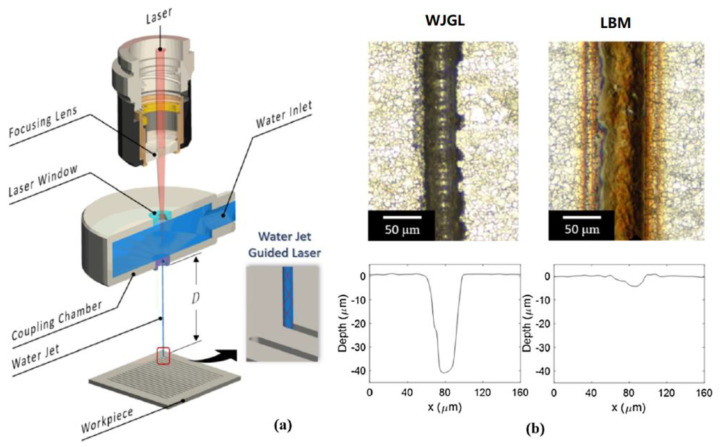
Schematic of the WJGL process and machined results; (**a**) schematic diagram of the WJGL process; (**b**) comparison of machined contour between the processes of WJGL and LBM [77].

**Figure 14 micromachines-12-01003-f014:**
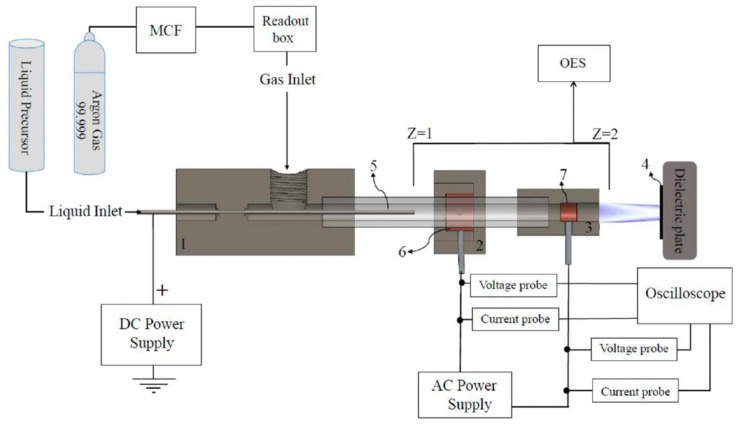
Schematic of a combination of electrospray and atmospheric pressure argon plasma jet [84].

**Figure 15 micromachines-12-01003-f015:**
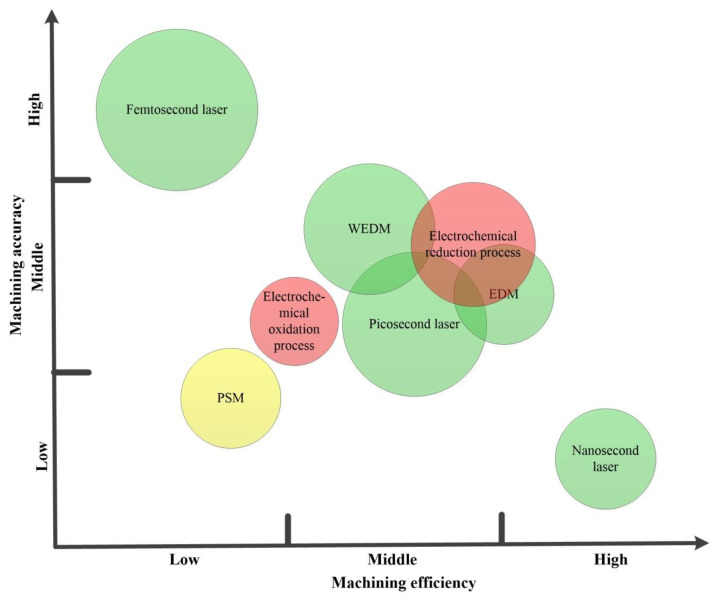
Comparison of machining efficiency, machining accuracy, machining cost, and environmental impact of different non-traditional machining methods for superhydrophobic surfaces. (The radius of the circle represents the machining cost; the larger the radius, the higher the cost. Moreover, the color of the circle represents the impact of manufacturing on the environment, while green, yellow, and red represent small, medium, and large impact on the environment).

**Table 1 micromachines-12-01003-t001:** Summary of the process of LBM described in the text grouped by laser type and object fabricated.

Laser Type	Object	Purpose	Findings	Remarks
Femtosecond laser	PMMA	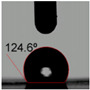 Inducing hydrophobic polymer surfaces	The WCA as a function of the energy deposition rate and the total energy deposited on the PMMA surface [14].	Different energies have an effect on the ratio of induced polar groups and non-polar groups, resulting in different surface hydrophobicity.
YSZ surfaces	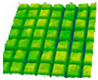 Fabrication of micro/nano composite structures	Laser texturing greatly improved the hydrophobicity of the YSZ surface, and the WCA increased from 86.7° to 151.8° [18].	Laser processing changes the ratio of polar groups to non-polar groups and ultimately improves the hydrophobicity of the surface.
Picosecond laser	Glass	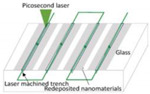 Fabricating multi-hierarchical micro/nano structures	A multi-level micro/nano structure could be fabricated in one step on any glass surface [21].	Quickly and effectively fabricate a liquid injection surface on glass materials.
Nanosecond laser	AISI 304	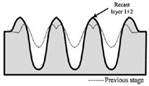 Fabrication of a micro-pin array with a high aspect ratio	The fabricated micro-pins array revealed adhesion on the vertical wall, and could produce 38.6 mN adhesion on the Ra 4.594 μm vertical wall [15].	The developed micro-pin array can be used in vertical surface attachment scenarios.
Cemented carbide	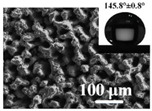 Obtaining a robust hydrophobic surface	The smaller the laser frequency, the smaller the decrease in the WCA; a relatively low scanning speed was more stable [16].	The hydrophobic surface fabricated by the proposed new method, liquid-phase laser ablation, has excellent abrasion resistance.
Aluminum alloy substrates	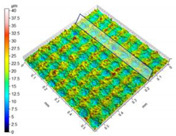 Fabrication of micro-pillar array	Experiments confirmed that the simulation using the proposed method was in good agreement with the results [17].	An accurate, low-cost and meaningful reference for the selection of functional surface manufacturing methods can be provided.
316 L stainless	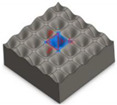 Obtaining Gaussian micro-holes array	They established a geometric model for laser processing of Gaussian micro-holes array [19].	The proposed model can optimize the geometric parameters of the micro-structure to achieve the maximum hydrophobicity.
Pyrolytic carbon	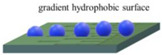 Fabricating a gradient hydrophobic surface with parallel ridges	A GHS was formed on the PYC [20].	The proposed method helps to design the PYC for artificial heart valve with good blood compatibility.

**Table 2 micromachines-12-01003-t002:** Summary of the process of EDM described in the text grouped by machining type and object fabricated.

Fabrication	Object	Purpose	Findings	Remarks
WEDM	AISI 304	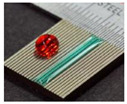 To fabricate grooves for controlling the wettability	A commercial control system was successfully applied to manufacture a micro-grooved wettability-controlled surface [22].	There is no need for any additional chemical treatment on the surface, which is conducive to low-cost industrial application.
	SiCp/Al composite	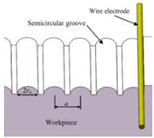 Fabrication of the wear-resistant superhydrophobic structure	Without other surface modification treatments, the maximum contact angle of the prepared micro-structure could reach 153.3° [25].	Through the proposed one-step machining method, the corrosion resistance, and anti-icing, self-cleaning, and other properties of Al-based composites can be improved at an economic cost.
	Copper	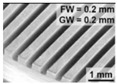 Obtaining a superhydrophobic hierarchical microgroove with nanocone structure	Compared with the flat hydrophobic copper, the CHT coefficient was increased by 82.9% [26].	The high-efficiency CHT interface developed, using copper micro/nano micro-machining technology, can have important application scope in the next generation of high heat dissipation in small spaces.
EDM	Pure copper	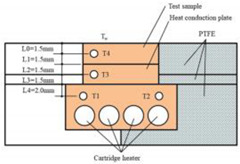 Fabrication of hierarchical micro/nano structure for heat transfer surface	The contact angle became larger as the discharge current increased [23].	The HMNS sample machined by the EDM process is hydrophobic, which increases the frequency of air bubbles falling off, resulting in a high heat transfer coefficient of the heat transfer surface.
	Copper sheet	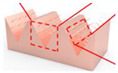 Fabricating nanowires on V-shaped micro-grooves	They proposed a simple and scalable method for preparing CuO nanowires with V-shaped micro-grooves [24].	A low-cost manufacturing micro-slot radiator is expected to be applied in energy, micro-electronics, chemical, medical, and other fields.
	6060 aluminum alloy	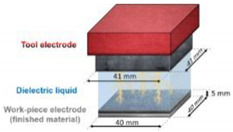 To investigate the relationship between wettability and surface micro-geometry	After EDM machining Al6060, there was a strong correlation between the surface morphology and the WCA [28].	The increase in roughness will also lead to a larger WCA, which may be a good index of wetting performance for the process of EDM.
MAE-EDM	Metal mesh	 Fabrication of a superhydrophobic mesh	The processed textured mesh had good contact angle stability, corrosion resistance and mechanical stability [27].	The fabric mesh not only provides universal separation of various oils, but also provides high separation rate and stable performance after recycling.

**Table 3 micromachines-12-01003-t003:** Summary of electrochemical oxidation process described in the text grouped by materials fabricated.

Materials	Authors, Year	Purpose	Findings	Remarks
Aluminum	Feng et al., 2002 [42]	Obtain poly(vinyl alcohol) nanofibers	A superhydrophobic surface with the WCA of 171.2° could be achieved	For the first time, amphiphilic materials are used to fabricate superhydrophobic surfaces. This method can be used to fabricate superhydrophobic surfaces from a variety of materials in the future.
	Zhang et al., 2006 [43].	Fabricate a two-dimensional array of perfluoropolyether derivative nanopillars	The nanopillars on the lotus leaf-like topology with low contact angles could be achieved.	The surface of two-dimensional array exhibit superhydrophobicity and self-cleaning properties.
	Neto et al., 2009 [44].	Obtain a surface formed by densely arranged nickel nanowires	The nickel nanowires had a very high aspect ratio.	It can be applied to the switchable wettability surface of micro-fluidic chips.
	Savio et al., 2021 [45].	Fabricate raised and recessed binary microstructures	The contact angle was indeed inversely proportional to the content of surface metal aluminum.	This study can better understand the formation mechanism of its superhydrophobic surface.
Silicon	Wang et al., 2007 [46].	Produce micro/nano topography with fractal shapes	Superhydrophobic silicon surfaces with the WCA of 160° could be fabricated.	This economical processing method has potential application prospects in technical fields such as electronic chip moisture prevention.
	Balucani et al., 2011 [47].	Making superhydrophobic surfaces with porous silicon	Large contact angles were observed on these surfaces, which had superhydrophobic properties.	This technology can provide a cheap and effective method for reducing friction in micro-fluidic applications.
Titanium	Wang et al., 2010 [48].	Create superhydrophobic surfaces on titanium materials	The experimental results showed that the control of wettability could be achieved on these surfaces.	It can be used in the application of oil sealing and anti-leakage on the engineering surface.
Copper	Wu et al., 2006 [49].	Fabricate stable superhydrophobic surfaces	The contact angle of the modified nano-needle surface was greater than 150°, and the inclination angle was less than 2°.	This simple and economical technology is expected to be applied to the walking parts of biomimetic robots with superhydrophobic sub-micron-fiber coating.
Others	He et al., 2010 [50].	Fabricate a ZnO film on zinc foil with a variety of structures	The surface of the ZnO film with electro wettability changed from a hydrophilic state to a superhydrophobic state.	The method provides a simple and rapid process for large-scale synthesis of different ZnO nanostructures, and adjusts the wettability of the ZnO nanostructures through an electric field.
	Shu et al., 2011 [51].	Make a superhydrophobic surface with a hierarchical structure	The surface was superhydrophobic, with a WCA of up to 175°.	Since the cone diameter and tip angle of the micro-capsules can controlled by anodizing conditions, there is a good application prospect.

**Table 4 micromachines-12-01003-t004:** Summary of electrochemical reduction process described in the text grouped by materials fabricated.

Materials	Authors, Year	Purpose	Findings	Remarks
Gold	Ye et al., 2010 [52]	Deposit gold microstructures on the ITO substrate	It had significant superhydrophobicity even in corrosive solutions with a wide pH range.	The fabricated surface has higher electro catalytic activity and stability to the electro oxidation of ethanol.
	Shi et al., 2005 [53].	Produce superhydrophobic coatings on gold wires	A small amount of micro/nano gold aggregates were formed on the surface of gold wire. The superhydrophobic coating with micro/nano gold aggregates could provide more bending force.	This understanding opens up new application prospects for bionic drag reduction and rapid advancement technology.
	Ren et al., 2009 [54].	Obtain hierarchical cauliflower-like gold structures	The surface of the cauliflower-like structure presented a high contact angle (WCA = 161.9°) and a low sliding angle.	The proposed two-step method for supporting electrochemical structures of gold micro/nano structures could be used in the ITO/glass substrates.
	Shepherdet al., 2020 [55].	Obtain heterogeneously mixed monolayers on the surface of polycrystalline gold	A local hydrophobic environment was formed near the molecular membrane.	It is expected to be used in the fields of anti-oxidation or anti-corrosion, chemical/biochemical sensors, etc. in the future.
Silver	Zhao et al., 2005 [56].	Making Ag aggregate dendritic structure on the polyelectrolyte multilayer film substrate	The prepared WCA was as high as 154°, and it became superhydrophobic.	The electrochemical deposition technology is used to control the density and morphology of the silver aggregates deposited on the multilayer film, which provides a possible new method for manufacturing self-cleaning surfaces.
	Gu et al., 2008 [57].	Grow single crystal Ag dendrites on Ni/Cu substrates	A superhydrophobic surface with a WCA of 154.5°+/−1.0° and an inclination angle of less than 2° could be obtained.	This method does not require a template and is simple and practical. Therefore, This self-cleaning surface has potential applications in nanotechnology.
Copper and copper oxides	Huang et al., 2011 [58].	Fabricate a particulate superhydrophobic aluminum surface	The aluminum substrate had a superhydrophobic surface with a roughness of 6–7 μm (WCA = 157°).	Nanostructured superhydrophobic aluminum surfaces can be prepared by two step processes: electrochemical deposition and electrochemical modification.
Si and Ag +	Yang et al., 2011 [59].	Fabricate a silicon micro/nano layered structure	By adjusting the process parameters, the morphology of the nanostructures could be partially controlled.	The superhydrophobic silicon surface produced by the ECM method has broad application prospects in micro/nano electromechanical systems (MEMS/NEMS).
Others	Huang et al., 2011 [60].	Deposit composite coating with a thorn-like hierarchical structure with high roughness	The geometry of this hierarchical structure could be controlled to make the contact angle as high as 174.9°.	Because this method saves time and money, it has broad application prospects in the industrial field.
	Xue et al., 2019 [61].	Fabricate a bimetallic NiCo coating with a layered micro-sphere structure on a carbon steel substrate	The layered micro-sphere structure of NiCo coating had a very high contact angle (about 165°) and exhibited superhydrophobic properties.	It has good anti-corrosion performance for bare carbon steel.
	Wang et al., 2020 [62].	Fabricate superhydrophobic cobalt-nickel coatings reinforced by micro/nano tungsten carbide (WC) particles	The prepared superhydrophobic Co-Ni/WC composite coating (with a WC content of 9.8 wt %) had excellent wear resistance.	The prepared Co-Ni/WC superhydrophobic coating with good mechanical durability is a promising alternative technology for corrosion protection.

**Table 5 micromachines-12-01003-t005:** Relative comparisons of different non-traditional machining processes for fabricating superhydrophobic surfaces.

Machining Process	Economy and Performances					
	Capital Investment	Tooling/Fixtures	Power Requirements	Tool Wear	Machining Efficiency	Machining Quality
LBM	Medium /High	Low	Middle	Very low	Low/Medium /High	Low/Medium/High
EDM	Medium	High	Low	High	Medium	Medium
ECM	Very high	Medium	High	Very low	Medium	Medium
USM	Low	Low	Low	Low	/	/
WJM	Medium	Medium	Middle	Low	/	/
PSM	High	Medium	High	Low	High	Low

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
