# Peer review of "Progress in Non-Traditional Processing for Fabricating Superhydrophobic Surfaces"

_micromachines, 2021, doi:10.3390/mi12091003_

Round 1
Reviewer 1 Report
This review provides an update on methods to fabricate superhydrophobic metal surfaces. The comparison between those methods is quite helpful to understand why a certain type of processing method is required. Considering the many updated literature is reviewed, I think this is a good review. In the meanwhile, I found some sentences are difficult to understand. Maybe the authors could further polish the English writing. I listed a few comments below for the authors to consider.
- page 1, line 23: the machining performances of the superhydrophobic surface...probably the author means the matching performances to fabricate the superhydrophobic surfaces. By the way, I thought "superhydrophobic" should be one word, which the author probably could have a double check.
- page 1, line 43: the reason for the superhydrophobicity
- page 2, line 50: wetting inherent in droplets... the local contact angle.. as ideal as possible. difficult to understand
- page 4, line 85: could the author provide the full name of USS
- line 88: what is salt spray corrosion?
- page 4, line 105: maybe change to "fabrication of materials with super..."
- page 9, line 225: what does it mean by nanosecond lasers are lower than..? probably you mean the timescale is much shorter or the price is cheaper?
- caption of figure 9: what do Ra and Rz mean?
- I suggest the author could carefully look at the manuscript to make more improvements than I listed above. That will be very helpful to improve the quality of the paper.
Author Response
Response to Reviewer 1 Comments
This review provides an update on methods to fabricate superhydrophobic metal surfaces. The comparison between those methods is quite helpful to understand why a certain type of processing method is required. Considering the many updated literature is reviewed, I think this is a good review. In the meanwhile, I found some sentences are difficult to understand. Maybe the authors could further polish the English writing. I listed a few comments below for the authors to consider.
- page 1, line 23: the machining performances of the superhydrophobic surface...probably the author means the matching performances to fabricate the superhydrophobic surfaces. By the way, I thought "superhydrophobic" should be one word, which the author probably could have a double check.
Reply: First of all, sincerely thank you for taking your time to review this article and thank you very much for your positive comments. Thank you very much for your suggestion. We have revised the errors in red color.
Revised Text:
“Third, the machining performances to fabricate the superhydrophobic surfaces by the electro-chemical machining (ECM), including electrochemical oxidation process and electrochemical reduction process, are reviewed grouped by materials fabricated.”
- page 1, line 43: the reason for the superhydrophobicity.
Reply: Thank you very much for your advices and comments. We have revised the errors in red color.
Revised Text:
“The reason for the superhydrophobicity is the special micro-scale topological structure presented on the surface.”
- page 2, line 50: wetting inherent in droplets... the local contact angle.. as ideal as possible. difficult to understand
Reply: Thank you very much for your advices and comments. In order to increase the readability of this review, we have deleted that sentence.
Revised Text:
“Figure 2 illustrates the typical wetting inherent in droplets (less than 0.5μl) placed on pigeon feathers [2]. The secret that …”
- page 4, line 85: could the author provide the full name of USS
Reply: Thank you very much for your advices and comments. The full name of USS is United States Ship, we have added it in red color.
Revised Text:
“The United States Ship (USS) McFall destroyer of the United States Navy has used protective outerwear made of superhydrophobic materials.”
- line 88: what is salt spray corrosion?
Reply: Thank you very much for your advices and comments. Salt spray corrosion is the oxidation reaction of chloride ions contained in the chloride salt (the main component is NaCl) with the metal surface, which damages the surface structure of the metal and reduces the strength of the metal. We have added it in red color.
Revised Text:
“This coat will protect the ship’s sensors, weapon systems, and other exposed equipment from salt spray corrosion. Among them, salt spray corrosion is the consequence of the oxidation reaction between the chloride ions contained in the chloride salt (the main com-ponent is NaCl) and the metal surface, which damages the surface structure of the metal and reduces the strength of the metal. This can significantly save time and money spent on ship maintenance [12].”
- page 4, line 105: maybe change to "fabrication of materials with super..."
Reply: Thank you very much for your advices and comments. We have revised it in red color.
Revised Text:
“Fabrication of materials with superhydrophobic have relatively high surface energy and exhibit obvious hydrophilicity, it is relatively more difficult to prepare superhydrophobic surfaces with metal as a matrix than glass and low surface energy polymers.”
- page 9, line 225: what does it mean by nanosecond lasers are lower than..? probably you mean the timescale is much shorter or the price is cheaper?
Reply: Thank you very much for your advices and comments. We have revised it in red color.
Revised Text:
“This is because the price of nanosecond lasers are cheaper than femtosecond lasers or picosecond lasers, and the timescale of nanosecond lasers are shorter than femtosecond lasers or picosecond lasers..”
- caption of figure 9: what do Ra and Rz mean?
Reply: Thank you very much for your advices and comments. Surface roughness Ra is the arithmetic mean deviation of the surface profile, the arithmetic mean of the absolute value of the profile deviation within the sampling length. Surface roughness Rz is the ten-point height of microscopic unevenness, the sum of the average of the five largest profile peak heights and the average of the five largest profile valley depths within the sampling length. We have revised it in red color.
Revised Text:
“Figure 9 depicted the relationship between the roughness (Ra and Rz) and the WCA of samples machined by the process of EDM. Among them, surface roughness Ra is the arithmetic mean deviation of the surface profile, the arithmetic mean of the absolute value of the profile deviation within the sampling length, and surface roughness Rz is the ten-point height of microscopic unevenness, the sum of the average of the five largest pro-file peak heights and the average of the five largest profile valley depths within the sampling length. As shown in Fig.9, with the increase of discharge current, more micro/Nano structures were processed, resulting in an increase in roughness (Ra and Rz).”
- I suggest the author could carefully look at the manuscript to make more improvements than I listed above. That will be very helpful to improve the quality of the paper.
Reply: Thank you very much for your advices and comments. We have improved it, sincerely thank you for taking your time to review this article and thank you very much for your positive comments.
I would like to resubmit this manuscript to “Micromachines”, and hope it is acceptable for publication in the journal. If there are any problems or questions about our paper, please do not hesitate to let us know.
Thank you very much for your attention to our paper.
Sincerely yours,
Xinggui Ren and Wuyi Ming

Reviewer 2 Report
The reviewers congratulate the authors for the submitted manuscript on an interesting topic. A well-structured paper is presented, where the following points should be reworked.
General notes:
- Is a military motivation required for this manuscript or are there similar civilian applications that could be referenced?
- The part of the sentence in line 108/109 “… and hopes to provide a certain reference for future research in related field.” can be deleted because it does not add any value to the manuscript.
- It is recommended to use the term water contact angle WCA every time the contact angle θ is measured on water to increase understanding and readability. In text sometimes, it is just called “contact angle θ”.
- There are recent references to the topic that have not been mentioned in the manuscript
- E.g., Kliuev, M. & Wiessner, M. & Büttner, H. & Maradia, U. & Wegener, Konrad. (2020). Super-hydrophobic and Super-hydrophilic Effect by Means of EDM Surface Structuring of γ-TiAl. Procedia CIRP. 95. 393-398. 10.1016/j.procir.2020.02.332.
- The different electrodes in line 242 are not named correctly. It is suggested to use the terms “tool electrode” and “workpiece electrode”.
- The last sentence of section 3.1 should be integrated at the beginning of the paragraph.
- In line 284, in the reference of Deng et Al. the connection to hydrophobic surfaces is not given like in previous references. Exemplary water contact angles WCA or similar should be presented to make the enhancement comparable to other references.
- In section “references”, the format should be consistent in terms of capitalization of words or whole phrases.
Language and typos:
- Throughout the whole document, line breaks and spelling should be rechecked.
- Terms containing the word micro should be spelled consistently.
- E.g., microstructures, micro structures, micro-structured, micro-pin, micro grooved
- Values and units should be written in a uniform way with or without space in text and figures.
- Throughout the document, the word "nano" is capitalized, although in most cases it should not.
- In line 321, it should be written “MAE-EDM” instead of “MAE-EDN”.
- The first sentence in line 339 should be rechecked regarding readability.
- The reviewer suggests introducing the abbreviation PSW in line 650.
- In line 730/731, the text font of the start of the sentence should be rechecked.
Figures:
- The way of presenting keywords in figures should be realised in a uniform way regarding the use of capital letters or similar.
- E.g., Figure 12, Figure 14
- The reference of figures in text should be consistent.
- E.g. Figure 2, Figure 4, Fig.5, Fig.3(a), Fig.3 (b)
- Scales should be added in figures in a uniform way to increase understanding and readability.
- The format for specifying units in chart labels should be consistent.
- E.g., Figure 7 and Figure 9
- In Figure 3, the reference is missing in the caption.
- In Figure 4, the unit “J/cm2/s” is not distinct and should be formulated clearly.
- In Figure 6 and Figure 7, different wording was used for the terms "abrasion" and "abrade" for the same meaning. Also, in line 167 the term “ablation” was used.
- In the caption of Figure 7, the unit “kHz” after “at 2mm/s” should be deleted.
- In Figure 11, it is suggested
- to insert “test sample (3)” above the images of the test samples instead of only writing “(3)” to increase understanding.
- to enlarge the angle representation to improve readability.
- to change the color of the angle´s values to increase readability on the white background.
- (a), (b), (c), (d) should be addressed in the caption.
- In Figure 10, it is recommended to increase text size for an increased readability.
- In Figure 15, the explanation of colors should be inserted in the scheme.
Tables:
- It is recommended to increase space between columns to increase the readability of close text blocks.
- In Table 1, the caption “Fabrication of micro-pillar array” is not added correctly. Also References of the images are missing.
- In Table 2, it is suggested to write at the first caption “Fabrication”.
- In Table 3, the space between the lines should be increased so that letters and lines do not overlap. Also, on the reference “Savio et al.”, the following comma is missing.
- In Table 4, the line after the row “others” and over Wang et al. should be rechecked.
- The space between text block and table should be increased.
- E.g., Table 4, Table 5
Author Response
Response to Reviewer 2 Comments
The reviewers congratulate the authors for the submitted manuscript on an interesting topic. A well-structured paper is presented, where the following points should be reworked.
General notes:
- Is a military motivation required for this manuscript or are there similar civilian applications that could be referenced?
Reply: First of all, sincerely thank you for taking your time to review this article and thank you very much for your positive comments. We have no military motives for this manuscript. Therefore, the main part of this review focuses on civil applications.
- The part of the sentence in line 108/109 “… and hopes to provide a certain reference for future research in related field.” can be deleted because it does not add any value to the manuscript.
Reply: Thank you very much for your advices and comments. We have deleted it.
Revised Text:
“Therefore, this study reviews and summarizes the development trends in the fabrication of superhydrophobic surface materials by non-traditional processing techniques, such as laser beam machining (LBM), electrical discharge machining (EDM), electrochemical machining (ECM), ultrasonic machining (USM), water jet machining (WJM), plasma spraying machining (PSM), etc.”
- It is recommended to use the term water contact angle WCA every time the contact angle θ is measured on water to increase understanding and readability. In text sometimes, it is just called “contact angle θ”.
Reply: Thank you very much for your advices and comments. We have revised it in red.
Revised Text:
“Superhydrophobic surface refers to the surface of such a material or object that can make the water droplets falling on it exhibit a water contact angle (WCA) greater than 150° [1].”
“Young first established the Young’s model [8], which believed that the surface of the material is smooth (ideally), and the WCA between the liquid droplet (usually a water droplet) and the smooth material surface is constant, as shown in Fig.3(a), and its size depends on the surface free energy..”
…
- There are recent references to the topic that have not been mentioned in the manuscript
E.g., Kliuev, M. & Wiessner, M. & Büttner, H. & Maradia, U. & Wegener, Konrad. (2020). Super-hydrophobic and Super-hydrophilic Effect by Means of EDM Surface Structuring of γ-TiAl. Procedia CIRP. 95. 393-398. 10.1016/j.procir.2020.02.332.
Reply: Thank you very much for your advices and comments. We have referenced it.
Revised Text:
“Kliuev et al. [89] compared the microstructure of the γ-TiAl surface processed by WEDM and die-sinking EDM, and found that these processes can be used as industrial solutions for the generation of superhydrophobic and superhydrophilic surfaces”
“89. Kliuev, M.; Wiessner, M.; Büttner, H.; et al. Super-hydrophobic and super-hydrophilic effect by means of edm surface struc-turing of γ-tial. Procedia CIRP, 2020, 95, pp. 393-398. DOI: 10.1016/j.procir.2020.02.332”
- The different electrodes in line 242 are not named correctly. It is suggested to use the terms “tool electrode” and “workpiece electrode”.
Reply: Thank you very much for your advices and comments. This is our mistake, we have revised it in red.
Revised Text:
“Electric discharge machining (EDM) uses the instantaneous high temperature generated when the spark between the tool electrode and the workpiece electrode is energized to remove the material on the surface of the workpiece.”
- The last sentence of section 3.1 should be integrated at the beginning of the paragraph.
Reply: Thank you very much for your advices and comments. We have revised it in red.
Revised Text:
“In the electric discharge machining (EDM) process, the metal is melted at a high temperature under the action of the discharge current between the tool electrode and the surface, and then solidifies on the surface to form a micro/Nano structure. EDM uses the instantaneous…”
- In line 284, in the reference of Deng et Al. the connection to hydrophobic surfaces is not given like in previous references. Exemplary water contact angles WCA or similar should be presented to make the enhancement comparable to other references.
Reply: Thank you very much for your advices and comments. We have added the maximum value of WCA in red.
Revised Text:
“All nanowire samples, which were fabricated by the proposed method, exhibited hydrophobicity (Maximum WCA with 140°).”
- In section “references”, the format should be consistent in terms of capitalization of words or whole phrases.
Reply: Thank you very much for your advices and comments. We have revised it in red.
Revised Text:
“6. Ball, P. Engineering shark skin and other solutions. Nature, 1999, 400(6744), pp. 507-509. DOI: 10.1038/22883
- Young, T. An Essay on the cohesion of fluids. Philosophical Transactions of the Royal Society of London, 1805, 95:65-87. DOI: 10.1098/rstl.1805.0005
- Neinhuis, C.; Barthlott, W. Characterization and distribution of water-repellent, self-cleaning plant surfaces. Annals of Botany, 1997, 79(6), pp. 667-677. DOI: 10.1006/anbo.1997.0400
- Wenzel; Robert, N.; Resistance of Solid Surfaces to Wetting by water. Transactions of the Faraday Society, 1936, 28(8), pp. 988-994. DOI: 10.1021/ie50320a024
- Mohri, N.; Fukuzawa, Y. Machining of insulating ceramics by EDM. Ceramics Japan, 2002, 37(1), pp. 35-51.
- Gotoh, H.; Tani, T.; Mohri, N. EDM of insulating ceramics by electrical conductive surface layer control. Procedia CIRP, 2016, 42, pp. 201-205. DOI: 10.1016/j.procir.2016.02.271
- Mcgeough, J. A. Principles of electrochemical machining. 1974. DOI: 10.1021/bk-1989-0390.ch039
- Feng, L.; Song, Y.; Zhai, J.; et al. Creation of a superhydrophobic surface from an amphiphilic polymer. Angewandte Chemie International Edition, 2003, 42(7), pp. 800-802. 10.1002/anie.200390212
- Ye, W.; Yan, J.; Ye, Q.; et al. Template-free and direct electrochemical deposition of hierarchical dendritic gold microstructures: growth and their multiple applications. Journal of Physical Chemistry C, 2010, 114(37), pp. 15617-15624. DOI: 10.1021/jp105929b
- Shi, F.; Wang, Z.; Zhang, X. Combining a layer‐by‐layer assembling technique with electrochemical deposition of gold ag-gregates to mimic the legs of water striders. Advanced Materials, 2005, 17(8), pp. 1005-1009. DOI: 10.1002/adma.200402090
- Huang, Y.; Sarkar, D.K.; Chen, X.G. Fabrication of sup erhydrophobic surfaces on aluminum alloy via electro dep osition of copp er followed by electro chemical modification. Nano-Micro Lett, 2011, 3(3), pp. 160-165. DOI: 10.1007/BF03353667
- Acherjee, B.; Maity, D.; Kuar, A.S. Ultrasonic machining process optimization by cuckoo search and chicken swarm optimi-zation algorithms. International Journal of Applied Metaheuristic Computing, 2020, 11(2), pp. 1-26. DOI: 10.4018/IJAMC.2020040101
- Kovacevic, R.; Hashish, M.; Mohan, R.; et al. State of the art of research and development in abrasive waterjet machining. Journal of Manufacturing Science & Engineering, 1997, 119(4), pp. 776-785. DOI: 10.1115/1.2836824
- Ramulu, M.; Kunaporn, S.; Arola, D.; et al. Waterjet machining and peening of metals. Journal of Pressure Vessel Technology, 1999, 122(1), pp. 90-95. DOI: 10.1115/1.556155
- Barletta, M.; Rubino, G.; Guarino, S.; et al. Fast regime-fluidized bed machining (FR-FBM) of atmospheric plasma spraying (APS) TiO2 coatings. Surface & Coatings Technology, 2008, 203(5-7), pp. 855-861. DOI: 10.1016/j.surfcoat.2008.05.043
- Barletta, M.; Rubino, G.; Guarino, S.; et al. Fast regime-fluidized bed machining (FR-FBM) of atmospheric plasma spraying (APS) TiO2 coatings. Surface & Coatings Technology, 2008, 203(5-7), pp. 855-861. DOI: 10.1016/j.surfcoat.2008.05.043
- Park, T.S.; Adomako, N.K.; Ashong, A.N.; et al. Interfacial structure and physical properties of high-entropy oxide coatings prepared via atmospheric plasma spraying. Coatings, 2021, 11(7), pp. 755. DOI: 10.3390/coatings11070755
- Yao, Y.; Wang, B.; Wang, J.; et al. Chemical machining of zerodur material with atmospheric pressure plasma jet. CIRP Annals - Manufacturing Technology, 2010, 59(1), pp. 337-340. DOI: 0.1016/j.cirp.2010.03.118
- Pandey, A.; Shan modern machining processes McGraw Hill, New Delhi ,1980, ISBN:978-93-8676-183-5
- Jing, Q.; Zhang, Y.; Kong, L.; et al. An investigation into accumulative difference mechanism in time and space for material removal in micro-EDM milling. Micromachines 2021, 12(6), pp. 711. DOI: 10.3390/mi12060711
- Quarto, M.; D’Urso, G.; Giardini, C.; et al. A Comparison between finite element model (FEM) sim-ulation and an integrated artificial neural network (ANN)-particle swarm optimization (PSO) approach to forecast per-formances of micro electro discharge Machining (Micro-EDM) Drilling. Micromachines, 2021, 12, pp. 667. DOI: 10.3390/mi12060667
- Tsai, H.Y.; Hsu, C.N.; Li, C.; et al. Surface wettability and electrical resistance analysis of droplets on Indium-Tin-Oxide glass fabricated using an ultraviolet laser system. Micromachines, 2021, 12(1), pp. 44. DOI: 10.3390/mi12010044
- Liu, J.; Liu, S. Dynamics behaviors of droplet on hydrophobic surfaces driven by electric field. Micromachines, 2019, 10(11), pp. 778.DOI: 10.3390/mi10110778”
Language and typos:
- Throughout the whole document, line breaks and spelling should be rechecked.
Reply: Thank you very much for your advices and comments. We have rechecked it.
- Terms containing the word micro should be spelled consistently.
E.g., microstructures, micro structures, micro-structured, micro-pin, micro grooved
Reply: Thank you very much for your advices and comments. We have revised it in red.
Revised Text:
“By changing the laser processing process parameters (energy density, scanning rate, scanning distance, laser frequency, etc.), the surface micro-structures with different mor-phology and size parameters can be obtained…”
“For laser processing of hydrophobic micro-structures, femtosecond lasers, picosecond lasers and nanosecond lasers have all been successfully applied [14-21]…”
“…”
- Values and units should be written in a uniform way with or without space in text and figures.
Reply: Thank you very much for your advices and comments. We have revised it in red.
Revised Text:
“The needle-like bristles covering the surface of the legs of the water strider are about 50μm in length, and the diameter gradually changes from 2 to 3μm at the root to 100nm at the tip, forming an inclination angle of about 20° with the surface of the leg.”
“Utilizing the conductive properties of graphene, the material can be electrically heated to prevent ice or deicing at lower temperatures, and only need to apply a voltage of 12volts to make the material anti-icing at a low temperature of -51°C [13].”
” Superhydrophobic/oleophobic fabrics can provide soldiers with up to 24hours of frost protection at -20°C. ”
” At a lower laser fluence of 0.299J/cm2, WCA increased and exceeded 90° while the total deposited energy accumulated above 300 J/cm2. When the total energy increases to the range of 600-900J/cm2, the WCA exceeded 120° (the maximum was about 125°). When the total energy was greater than 900J/cm2, the WCA decreased with the increase of the total energy. In addition, Figures 4a and b also showed that the optimal energy deposition rate to achieve the maximum WCA was about 50J/cm2/s during a single scan of the laser beam.”
” Figure 6. The influence of laser frequency on the wettabilitys (SEM images) [16]; (a) The hydrophobic surface fabricat-ed at 20kHz before abrasion; (b) The hydrophobic surface fabricated at 160kHz before abrasion; (c) The hydrophobic surface fabricated at 20kHz after 600 abrasion cycles; (d) The hydrophobic surface fabricated at 160kHz after 600 abra-sion cycles. ”
” In addition, when it heated in the air for more than 42hours (over 160°C), the surface of the copper still maintained its superhydrophobic properties.”
- Throughout the document, the word "nano" is capitalized, although in most cases it should not.
Reply: Thank you very much for your advices and comments. We have revised it in red.
Revised Text:
“In addition, there are fine semi-helical nano-grooves on the surface of each bristles.”
” In this way, a multi-level micro/nano structure could be fabricated in one step on any glass surface.”
”…”
13.In line 321, it should be written “MAE-EDM” instead of “MAE-EDN”.
Reply: Thank you very much for your advices and comments. This is our mistake; we have revised it in red.
Revised Text:
“Compared with the flat hydrophobic copper, the CHT coefficient was increased by 82.9%. Based on magnetically aided electrode EDM (MAE-EDM), Xiao et al. [27] fabricated a superhydrophobic mesh.”
- The first sentence in line 339 should be rechecked regarding readability.
Reply: Thank you very much for your advices and comments. We have revised it for increase the readability.
Revised Text:
“According to the existing literatures, it is feasible for EDM to cut a hydrophobic micro-structured surface, such as grooves, micro/nano structure, and hydrophobic meshes, the processing steps of EDM are few, and it is convenient for large-scale industrial applications in the future.”
- The reviewer suggests introducing the abbreviation PSW in line 650.
Reply: Thank you very much for your advices and comments. This is our mistake, and we have revised it in red.
Revised Text:
“Other non-contact processes such as the USM and the PSM processes, also have their advantages and disadvantages as reported by different researchers [92].”
- In line 730/731, the text font of the start of the sentence should be rechecked.
Reply: Thank you very much for your advices and comments. we have revised it in red.
Revised Text:
“Acknowledgments: The authors acknowledge the financial support provided by the Local Inno-vative and Research Teams Project of Guangdong Pearl River Talents Program (2017BT01G167) and by the Guangdong Youth Talent Innovation Project (2019GKQNCX092).”
Figures:
- The way of presenting keywords in figures should be realised in a uniform way regarding the use of capital letters or similar.
E.g., Figure 12, Figure 14
Reply: Thank you very much for your advices and comments. We have changed Figure 12.
Revised Figure:
Figure 12. Schematic of ultrasonic vibration-assisted laser composite machining of copper for superhydrophobicity [70].
- The reference of figures in text should be consistent.
E.g. Figure 2, Figure 4, Fig.5, Fig.3(a), Fig.3 (b)
Reply: Thank you very much for your advices and comments. We have revised them in red, among them, Fig.3 (a), Fig.3 (b) and Fig.3 (c) were drew by ourselves.
Revised Text:
“Figure 2. The superhydrophobic surface of bird feathers [2]; (a) General image of water droplets placed on bird feathers; (b) SEM image of the central part of a bird feather.”
“Figure 4. Effect of energy deposition rate and total energy on the WCA at different laser fluence [14]; (a) 0.299 J/cm2; (b) 0.556 J/cm2.’’
’’ Figure 5. Micro-pins array fabricated on stainless steel sheet (AISI 304, 3 mm×3 mm) by the process of LBM [15]. ’’
- Scales should be added in figures in a uniform way to increase understanding and readability. The format for specifying units in chart labels should be consistent.
E.g., Figure 7 and Figure 9
Reply: Thank you very much for your advices and comments. Figure 7 and Figure 9 are original drawings, and due to copyright restrictions, it is inconvenient for us to modify them.
- In Figure 3, the reference is missing in the caption.
Reply: Thank you very much for your advices and comments. Fig.3(a), Fig.3 (b) and Fig.3(c) were drew by ourselves.
- In Figure 4, the unit “J/cm2/s” is not distinct and should be formulated clearly.
Reply: Thank you very much for your advices and comments. We have checked it. This is the original image. Due to copyright restrictions, it is inconvenient for us to make changes.
- In Figure 6 and Figure 7, different wording was used for the terms "abrasion" and "abrade" for the same meaning. Also, in line 167 the term “ablation” was used.
Reply: Thank you very much for your advices and comments. We have revised Figure 6, by the way, “ablation” meaning is different to "abrasion".
Revised Figure:
Figure 6. The influence of laser frequency on the wettabilitys (SEM images) [16]; (a) The hydrophobic surface fabricated at 20kHz before abrasion; (b) The hydrophobic surface fabricated at 160kHz before abrasion; (c) The hydrophobic surface fabricated at 20kHz after 600 abrasion cycles; (d) The hydrophobic surface fabricated at 160kHz after 600 abrasion cycles.
- In the caption of Figure 7, the unit “kHz” after “at 2mm/s” should be deleted.
Reply: Thank you very much for your advices and comments. This is our mistake, and we have revised it in red.
Revised Text:
“(c) The hydrophobic surface fabricated at 2mm/s after 600 abrasion cycles; (d) The hydrophobic surface fabricated at 64mm/s after 600 abrasion cycles.”
- In Figure 11, it is suggested:
to insert “test sample (3)” above the images of the test samples instead of only writing “(3)” to increase understanding.
to enlarge the angle representation to improve readability.
to change the color of the angle´s values to increase readability on the white background.
(a), (b), (c), (d) should be addressed in the caption.
Reply: Thank you very much for your advices and comments. We have revised Figure 11 and addressed the caption of (a), (b), (c), (d).
Revised Figure and Text:
Figure 11. Relationship between designed parameters of the groove and the contact state [25]. (a) The original sample (3); (b) The sample (3) with deep groove; (c) The original sample (4); (d) The sample (4) with deep groove.
- In Figure 10, it is recommended to increase text size for an increased readability.
Reply: Thank you very much for your advices and comments. Due to increase the readability, we have revised the Figure 10.
Revised Figure:
Figure 10. Schematic of the manufacturing process for the V-shaped microgrooves; (a) Geometric dimensions; (b) Schematic of the thermal oxidation process [24].
- In Figure 15, the explanation of colors should be inserted in the scheme.
Reply: Thank you very much for your advices and comments. We have explained the meaning of colors in review, and already marked in red.
Revised Text:
“Figure 15. Comparisons of machining efficiency, machining accuracy, machining cost and environmental impact of different non-traditional machining methods for superhydrophobic surfaces (The radius of the circle represents the machining cost; the larger the radius, the higher the cost. Moreover, the color of the circle represents the impact of manufacturing on the environment, while green, yellow and red represent small, medium and large impact on the environment).”
Tables:
- It is recommended to increase space between columns to increase the readability of close text blocks.
Reply: We have adjusted as much as possible. If our review paper is accepted, the publishing editor may make further adjustments.
- In Table 1, the caption “Fabrication of micro-pillar array” is not added correctly. Also References of the images are missing.
Reply: Thank you very much for your advices and comments. We have revised it in red.
Revised Tables:
Table 1. Summary of the process of LBM described in the text grouped by laser type and object fabricated.
Laser type |
Object |
Purpose |
Findings |
Remarks |
Femtosecond laser |
PMMA |
Inducing hydrophobic polymer surfaces |
The WCA as a function of the energy deposition rate and the total energy deposited on the PMMA surface [14]. |
Different energies have an effect on the ratio of induced polar groups and non-polar groups, resulting in different surface hydrophobicity. |
YSZ surfaces |
Fabrication of Micro/nano composite structures |
Laser texturing greatly improved the hydrophobicity of the YSZ surface, and the WCA increased from 86.7° to 151.8° [18]. |
Laser processing changes the ratio of polar groups to non-polar groups, and ultimately improves the hydrophobicity of the surface. |
|
Picosecond laser |
Glass |
Fabricating multi-hierarchical micro/nano structures |
A multi-level micro/nano structure could be fabricated in one step on any glass surface [21]. |
Quickly and effectively fabricate a liquid injection surface on glass materials. |
Nanosecond laser |
AISI 304 |
Fabrication of a micro-pin array with a high aspect ratio |
The fabricated micro-pins array revealed adhesion on the vertical wall, and could produce 38.6mN adhesion on the Ra 4.594μm vertical wall [15]. |
The developed micro-pin array can be used in vertical surface attachment scenarios. |
Cemented carbide |
Obtaining a robust hydropbic surface |
The smaller the laser frequency, the smaller the decrease in the WCA; a relatively low scanning speed was more stable [16]. |
The hydrophobic surface fabricated by the proposed new method, liquid-phase laser ablation, has excellent abrasion resistance. |
|
Aluminum alloy substrates |
Fabrication of micro-pillar array |
Experiments confirmed that the simulation using the proposed method was in good agreement with the results [17]. |
An accurate, low-cost and meaningful reference for the selection of functional surface manufacturing methods can be provided. |
|
316L stainless |
Obtaining Gaussian micro-holes array |
They established a geometric model for laser processing of Gaussian micro-holes array[19] |
The proposed model can optimize the geometric parameters of the micro-structure to achieve the maximum hydrophobicity |
|
Pyrolytic carbon |
Fabricating a gradient hydrophobic surface with parallel ridges |
A GHS was formed on the PYC [20]. |
The proposed method helps to design the PYC for artificial heart valve with good blood compatibility. |
- In Table 2, it is suggested to write at the first caption “Fabrication”.
Reply: Thank you very much for your advices and comments. We have revised it in red.
Revised Tables:
Table 2. Summary of the process of EDM described in the text grouped by machining type and object fabricated.
Fabrication |
Object |
Purpose |
Findings |
Remarks |
WEDM |
AISI 304 |
To fabricate grooves for controlling the wettability |
A commercial control system was successfully applied to manufacture a micro-grooved wettability-controlled surface [22]. |
There is no need for any additional chemical treatment on the surface, which is conducive to low-cost industrial application. |
SiCp/Al composite |
Fabrication of the wear-resistant superhydrophobic structure |
Without other surface modification treatments, the maximum contact angle of the prepared mcro-structure could reach 153.3° [25]. |
Through the proposed one-step machining method, the corrosion resistance, and anti-icing, self-cleaning and other properties of Al-based composites can be improved at an economic cost. |
|
Copper |
Obtaining a superhydrophobic hierarchical microgroove with nanocone structure |
Compared with the flat hydrophobic copper, the CHT coefficient was increased by 82.9% [26]. |
The high-efficiency CHT interface developed, using copper micro/nano machining technology, can have important application scope in the next generation of high heat dissipation in small spaces. |
|
EDM |
Pure copper |
Fabrication of hierarchical micro/Nano structure for heat transfer surface |
The contact angle became larger as the discharge current increased [23]. |
The HMNS sample machined by the EDM process is hydrophobic, which increases the frequency of air bubbles falling off, resulting in a high heat transfer coefficient of the heat transfer surface. |
Copper sheet |
Fabricating nanowires on V-shaped micro-grooves |
They proposed a simple and scalable method for preparing CuO nanowires with V-shaped micro-grooves [24]. |
A low-cost manufacturing micro-slot radiator is expected to be applied in energy, micro-electronics, chemical, medical and other fields. |
|
6060 aluminum alloy |
To investigate the relationship between wettability and surface micro-geometry |
After EDM machining Al6060, there was a strong correlation between the surface morphology and the WCA [28]. |
The increase in roughness will also lead to a larger WCA, which may be a good index of wetting performance for the process of EDM. |
|
MAE-EDM |
Metal mesh |
Fabrication of a superhydrophobic mesh |
The processed textured mesh had good contact angle stability, corrosion resistance and mechanical stability [27]. |
The fabric mesh not only provides universal separation of various oils, but also provides high separation rate and stable performance after recycling. |
- In Table 3, the space between the lines should be increased so that letters and lines do not overlap. Also, on the reference “Savio et al.”, the following comma is missing.
Reply: Thank you very much for your advices and comments. We have revised it in red.
Revised Tables:
Table 3. Summary of electrochemical oxidation process described in the text grouped by materials fabricated.
Materials |
Authors, Year |
Purpose |
Findings |
Remarks |
Aluminium |
Feng et al., 2002 [42] |
Obtain poly(vinyl alcohol) nanofibers |
A superhydrophobic surface with the WCA of 171.2° could be achieved |
For the first time, amphiphilic materials are used to fabricate superhydrophobic surfaces. This method can be used to fabricate superhydrophobic surfaces from a variety of materials in the future. |
Zhang et al., 2006 [43]. |
Fabricate a two-dimensional array of perfluoropolyether derivative nanopillars |
The nanopillars on the lotus leaf-like topology with low contact angles could be achieved. |
The surface of two-dimensional array exhibit superhydrophobicity and self-cleaning properties. |
|
Neto et al., 2009 [44]. |
Obtain a surface formed by densely arranged nickel nanowires |
The nickel nanowires had a very high aspect ratio. |
It can be applied to the switchable wettability surface of micro-fluidic chips. |
|
Savio et al., 2021 [45]. |
Fabricate raised and recessed binary microstructures |
The contact angle was indeed inversely proportional to the content of surface metal aluminum. |
This study can better understand the formation mechanism of its superhydrophobic surface. |
|
Silicon |
Wang et al., 2007 [46]. |
Produce micro/nano topography with fractal shapes |
Superhydrophobic silicon surfaces with the WCA of 160° could be fabricated. |
This economical processing method has potential application prospects in technical fields such as electronic chip moisture prevention. |
Balucani et al., 2011 [47]. |
Making superhydrophobic surfaces with porous silicon |
Large contact angles were observed on these surfaces, which had superhydrophobic properties. |
This technology can provide a cheap and effective method for reducing friction in micro-fluidic applications. |
|
Titanium |
Wang et al., 2010 [48]. |
Create superhydrophobic surfaces on titanium materials |
The experimental results showed that the control of wettability could be achieved on these surfaces. |
It can be used in the application of oil sealing and anti-leakage on the engineering surface. |
Copper |
Wu et al., 2006 [49]. |
Fabricate stable superhydrophobic surfaces |
The contact angle of the modified nano-needle surface was greater than 150°, and the inclination angle was less than 2°. |
This simple and economical technology is expected to be applied to the walking parts of biomimetic robots with superhydrophobic sub-micron-fiber coating. |
Others |
He et al., 2010 [50]. |
Fabricate a ZnO film on zinc foil with a variety of structures |
The surface of the ZnO film with electro wettability changed from a hydrophilic state to a superhydrophobic state. |
The method provides a simple and rapid process for large-scale synthesis of different ZnO nanostructures, and adjusts the wettability of the ZnO nanostructures through an electric field. |
Shu et al., 2011 [51]. |
Make a superhydrophobic surface with a hierarchical structure |
The surface was superhydrophobic, with a WCA of up to 175°. |
Since the cone diameter and tip angle of the micro-capsules can controlled by anodizing conditions, there is a good application prospect. |
- In Table 4, the line after the row “others” and over Wang et al. should be rechecked.
Reply: Thank you very much for your advices and comments. We have revised it in red.
Revised Tables:
Table 4. Summary of electrochemical reduction process described in the text grouped by materials fabricated.
Materials |
Authors, Year |
Purpose |
Findings |
Remarks |
Gold |
Ye et al., 2010 [52] |
Deposit gold microstructures on the ITO substrate |
It had significant superhydrophobicity even in corrosive solutions with a wide pH range. |
The fabricated surface has higher electro catalytic activity and stability to the electro oxidation of ethanol. |
Shi et al., 2005 [53]. |
Produce superhydrophobic coatings on gold wires |
A small amount of micro/nano gold aggregates were formed on the surface of gold wire. And, the superhydrophobic coating with micro/nano gold aggregates could provide more bending force. |
This understanding opens up new application prospects for bionic drag reduction and rapid advancement technology. |
|
Ren et al., 2009 [54]. |
Obtain hierarchical cauliflower-like gold structures |
The surface of the cauliflower-like structure presented a high contact angle (WCA=161.9°) and a low sliding angle. |
The proposed two-step method for supporting electrochemical structures of gold micro/nano structures could be used in the ITO/glass substrates. |
|
Shepherdet al., 2020 [55]. |
Obtain heterogeneously mixed monolayers on the surface of polycrystalline gold |
A local hydrophobic environment was formed near the molecular membrane |
It is expected to be used in the fields of anti-oxidation or anti-corrosion, chemical/biochemical sensors, etc. in the future. |
|
Silver |
Zhao et al., 2005 [56]. |
Making Ag aggregate dendritic structure on the polyelectrolyte multilayer film substrate |
The prepared WCA was as high as 154°, and it became superhydrophobic |
The electrochemical deposition technology is used to control the density and morphology of the silver aggregates deposited on the multilayer film, which provides a possible new method for manufacturing self-cleaning surfaces. |
Gu et al., 2008 [57]. |
Grow single crystal Ag dendrites on Ni/Cu substrates |
A superhydrophobic surface with a WCA of 154.5°+/-1.0° and an inclination angle of less than 2° could be obtained. |
This method does not require a template and is simple and practical. Therefore, This self-cleaning surface has potential applications in nanotechnology. |
|
Copper and copper oxides |
Huang et al., 2011 [58]. |
Fabricate a particulate superhydrophobic aluminum surface |
The aluminum substrate had a superhydrophobic surface with a roughness of 6-7μm (WCA=157°) |
Nanostructured superhydrophobic aluminum surfaces can be prepared by two step processes: electrochemical deposition and electrochemical modification. |
Si and Ag + |
Yang et al., 2011 [59]. |
Fabricate a silicon micro/nano layered structure |
By adjusting the process parameters, the morphology of the nanostructures could be partially controlled. |
The superhydrophobic silicon surface produced by the ECM method has broad application prospects in micro/nano electromechanical systems (MEMS/NEMS). |
Others |
Huang et al., 2011 [60]. |
Deposit composite coating with a thorn-like hierarchical structure with high roughness |
The geometry of this hierarchical structure could be controlled to make the contact angle as high as 174.9°. |
Because this method saves time and money, it has broad application prospects in the industrial field. |
Xue et al., 2019 [61]. |
Fabricate a bimetallic NiCo coating with a layered micro-sphere structure on a carbon steel substrate |
The layered micro-sphere structure of NiCo coating had a very high contact angle (about 165°) and exhibited superhydrophobic properties. |
It has good anti-corrosion performance for bare carbon steel. |
|
Wang et al., 2020 [62]. |
Fabricate superhydrophobic cobalt-nickel coatings reinforced by micro/nano tungsten carbide (WC) particles |
The prepared superhydrophobic Co-Ni/WC composite coating (with a WC content of 9.8wt %) had excellent wear resistance. |
The prepared Co-Ni/WC superhydrophobic coating with good mechanical durability is a promising alternative technology for corrosion protection. |
- The space between text block and table should be increased.
E.g., Table 4, Table 5
Reply: We have adjusted as much as possible. If our review paper is accepted, the publishing editor may make further adjustments.
I would like to resubmit this manuscript to “Micromachines”, and hope it is acceptable for publication in the journal. If there are any problems or questions about our paper, please do not hesitate to let us know.
Thank you very much for your attention to our paper.
Sincerely yours,
Xinggui Ren and Wuyi Ming
